



# Development of an atmosphere-ocean coupled operational forecast model for the Maritime Continent: Part 1 - Evaluation of ocean forecasts

Bijoy Thompson[1], Claudio Sanchez[2,3], Boon Chong Peter Heng[3], Rajesh Kumar[3], Jianyu Liu[3], Xiang-Yu Huang[3] and Pavel Tkalich[1]

[1]Tropical Marine Science Institute, National University of Singapore, Singapore 119222, Singapore.
[2]Met Office, Exeter, EX1 3PB, United Kingdom.
[3]Centre for Climate Research Singapore, Meteorological Service Singapore, Singapore 537054, Singapore

*Correspondence to*: Bijoy Thompson (bijoymet@email.com)

**Abstract.** This article describes the development and ocean forecast evaluation of an atmosphere-ocean coupled prediction system for the Maritime Continent (MC) domain, which includes the eastern Indian and western Pacific Oceans. The coupled system comprises regional configurations of the atmospheric model MetUM and ocean model NEMO, at a uniform horizontal resolution of 4.5 km x 4.5 km, coupled using the OASIS3-MCT libraries. The coupled model is run as a pre-operational forecast system from 1 to 31 October 2019. Hindcast simulations performed for the period 1 January 2014 to 30 September

2019, using the stand-alone ocean configuration, provided the initial condition to the coupled ocean model. This paper details the evaluations of ocean-only model hindcast and 6-day coupled ocean forecast simulations. Direct comparison of sea surface temperature (SST) and sea surface height (SSH) with analysis as well as in situ observations are performed for the ocean-only hindcast evaluation. For the evaluation of coupled ocean model, comparisons of ocean forecast for different forecast lead times with SST analysis, and in situ observations of SSH, temperature and salinity have been performed. Overall, the model forecast

deviation of SST, SSH, and subsurface temperature and salinity fields relative to observation is within acceptable error limits of operational forecast models. Typical runtimes of the daily forecast simulations are found to be suitable for the operational forecast applications.

## 1 Introduction

Dynamical processes and flux exchanges between the Earth system components are better represented in coupled modelling

systems rather than the single component models (e.g. Meehl, 1990). Hence, coupled models, particularly with dynamically interactive atmosphere, ocean, land surface and sea-ice models, are increasingly employed for climate research as well as operational forecast applications (e.g. Miller et al., 2017; Lewis et al., 2018, 2019a). The atmosphere and ocean are two major components of the Earth's climate system, and interactions between these two systems are key drivers of climate and weather. In the past, efforts toward the development of atmosphere-ocean coupled models were largely constrained by their high

computational requirements, limited understanding of air-sea coupled processes and lower computational efficiency (Meehl,





1990). During the last three decades, there have been significant advancements in the computational power of supercomputers and the computational efficiency of atmosphere/ocean circulation models. Presently, global atmosphere-ocean-wave-land surface-sea ice coupled operational forecasts are available at spatial resolutions of 0.1º in the Integrated Forecast Systems (IFS) developed by the European Centre for Medium Range Weather Forecasting (ECMWF) to 0.25º in the Global Forecast System

(GFS) developed by the National Center for Environmental Prediction (NCEP). Moreover, the accessibility of High-Performance Computers (HPC) to researchers has considerably increased in the last decade. Several regional and global atmosphere-ocean coupled modelling systems have been developed worldwide during this period (see reviews by Giorgi and Gutowsky, 2015 and Xue et al., 2020).

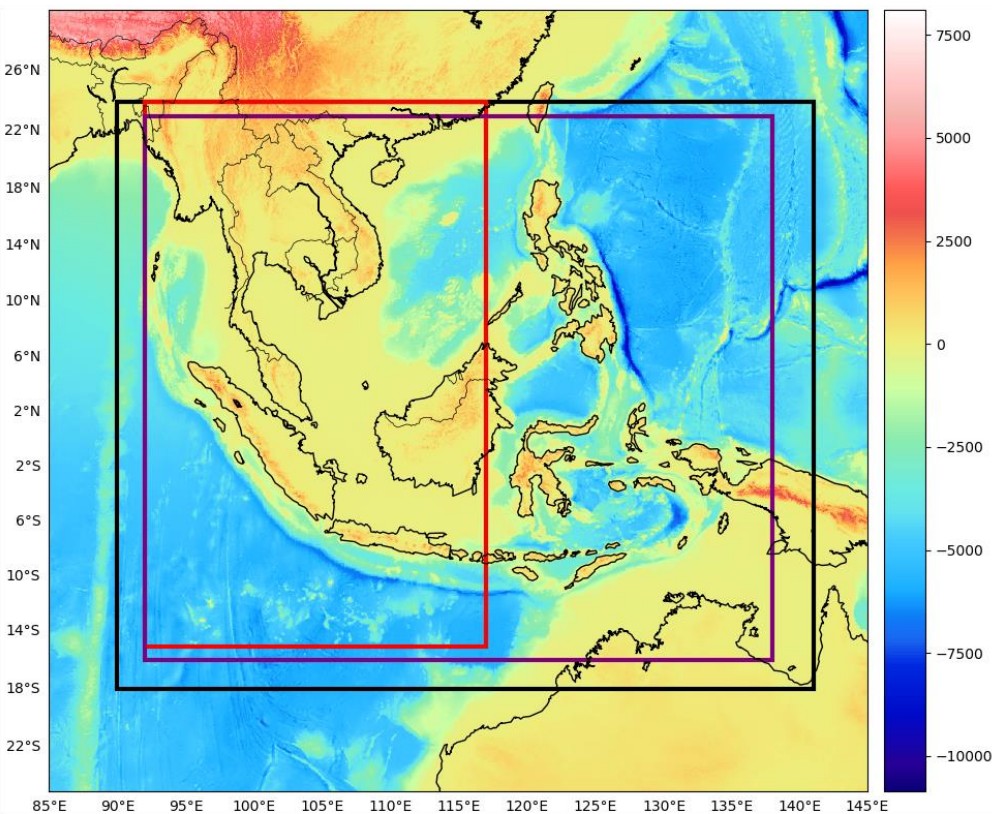

**Figure 1:** Bathymetry and orography (in m) of the Maritime Continent from GEBCO 2014 data. MC coupled model domain (black colour box), western Maritime Continent domain in Thompson et al. (2018) (red colour box) and domain used for ocean forecast evaluation in present study (purple colour box).





The tropical region lying between the eastern Indian Ocean and west Pacific Ocean, encompassing the Malay Peninsula, Philippine Islands, and Indonesian Archipelago and surrounding oceanic/island region is generally referred to as the Maritime Continent (MC). This region is characterized by complex orography and shallow seas interconnected by numerous straits (Figure 1). The MC region is characterized by strong atmosphere-ocean coupled processes across multiple timescales. The El Nino Southern Oscillation (Bjerknes, 1969) and the Indian Ocean Dipole/Zonal mode (Saji et al., 1999; Webster et al., 1999)

represent two dominant climate modes of variability that influences the MC in inter-annual timescales. Meanwhile, the monsoons and Madden Julian Oscillations (MJO, Madden and Julian, 1994) manifest the coupled processes over the MC in seasonal and intra-seasonal scales, respectively. Because of its geographical location in the middle of the Indo-Pacific warm pool and in the ascending branch of global atmospheric Walker circulation, the MC has been identified as an area of climatic importance both in regional and global environments (Neale and Slingo, 2003; Qu et al., 2005).

The development of regional coupled models is mainly driven by the idea that by resolving fine-scale orographic, ocean circulation and coastal ocean features, a more accurate representation of atmosphere/ocean dynamics and coupled processes can be achieved. The prediction of atmospheric and oceanic variables over the MC is challenging because of its complex geography, strong air-sea coupling and remote ocean influences. Earlier studies suggested that the accuracy of atmospheric and ocean hindcast/forecast significantly improves when the simulations are performed using coupled models (Xue et al.,

2014; Thompson et al., 2018; Lewis et al., 2019a). There have been a few coupled modelling studies over the MC focusing on the climate/weather research or short-range atmosphere-ocean forecasting (e.g. Aldrian et al., 2005; Wei et al., 2014; Li et al., 2017; Thompson et al., 2018). Recently, Xue et al. (2020) presented a review of atmosphere-ocean coupled modelling studies over the MC region.

Besides coupling, the model skill in simulating atmosphere and ocean state shows a strong relation to the grid resolution also

(e.g. Li et al., 2017). Local mesoscale processes (e.g. land/sea-breezes) also play an important role in the adequate simulation of upscale processes such as the MJO (Birch et al., 2015). Convection plays a fundamental role, either locally or embedded in bigger envelopes such as the MJO, influencing the diurnal cycle of precipitation and moving squall lines (Love et al., 2011). Therefore, the simulation of weather and climate processes over the MC requires sufficient resolution to resolve these scales and their interactions. Generally, a horizontal resolution of approx. 4 km, so-called convection permitting, has been effective

in representing fine-scale processes over the MC (Love et al., 2011; Birch et al., 2014, 2015; Vincent and Lane, 2017). The first attempt towards the development of a convection-permitting atmosphere-ocean coupled model over the MC was undertaken by Thompson et al. (2018, hereafter T18). T18 used a regional version of the UK Met Office Unified Model (MetUM) atmospheric model and Nucleus for European Modelling of the Ocean (NEMO) ocean model configured for the western MC (WMC). For simplicity, the WMC coupled model configuration used in T18 is referred to as WMCao hereafter.

The atmosphere and ocean components of WMCao were configured for the same domain and similar horizontal resolution of 4.5 km x 4.5 km (Figure 1). The model resolution is fine enough to represent the complex coastal geography, ocean bathymetry at shallow oceans and straits, and orographic features such as mountain ranges with enormous influences in local weather (e.g., Bukit barisan in Sumatra, Luzon Sierra).





Work to develop and first-hand evaluation of the WMCao, described in T18, was a preliminary step aimed to establish a high-
resolution atmosphere-ocean coupled model focusing on the Southeast Asia region for both operational forecast and climate
research applications. Since the overall objective of the development was to simulate both atmospheric and oceanic variables,
the coupling has provided a better consistency between the atmospheric conditions and that of the ocean underneath rather
than employing stand-alone models. The case studies conducted as part of the WMCao evaluation suggested that for an
accurate prediction of weather events such as cold surges or typhoons, the zonal extent of the domain might not be sufficient.
For instance, cyclogenesis inside the South China Sea (SCS) is rare and most of the cyclones/typhoons that appear over the
SCS are originated in the northwestern Pacific Ocean. The north Pacific Ocean region between 100ºE and 180º is the most
active tropical cyclone basin in the earth and it accounts for about one-third of the world tropical cyclone annually. Hence,
rather than internal coupled dynamics, the predicted track and typhoon characteristics are dominantly driven by the Lateral
Boundary Conditions (LBC) in WMCao. Similarly, the simulation of MJO and cold surge related weather parameters may
also be heavily influenced by LBC. Hence, to address the issues encountered in T18 and incorporate the latest model scientific
developments, the present study aims to bring several key upgrades to the WMCao configuration, and test its feasibility in
operational forecast application. Main updates to the coupled modelling system include extending the eastern boundary of the
model domain to the west Pacific Ocean, upgrading MetUM to the latest science configuration and incorporating tide boundary
forcing to the NEMO.

The study presents details of the atmosphere-ocean coupled prediction system developed for the MC and an evaluation of the
ocean forecast from the system using a 6-day pre-operational forecast for October 2019. Following the method employed in
many earlier coupled modelling studies (e.g. Li et al., 2014; Lewis et al., 2018), the evaluation of WMCao in T18 has been
performed by using short case study simulations, spanning over 5-days, of selected weather events. In the present study, instead
of case studies, we assess surface and subsurface oceanic variables predicted by the coupled system across different forecast
lead times.

The next section of this paper presents an overview of the model setup, including a brief description of the model domain,
atmospheric, ocean and coupled model configurations. A brief discussion of the pre-operational forecast system setup is also
presented. Section 3 provides the details of datasets used for the atmosphere/ocean model forcing and evaluation. Section 4
presents an assessment of the sea surface variables simulated by the stand-alone ocean model and both surface and subsurface
ocean forecast delivered by the MC coupled model. Finally, section 5 summarizes the results obtained from the study and
suggests future developments.

## 2 Model setup

### 2.1 Model domain

The model domain extends from 92ºE to 141ºE and 18ºS to 24ºN (Figure 1) on a regular latitude-longitude grid, that covers
most of the tropical regions of eastern Indian and western Pacific Oceans. The deepest oceanic trench on the Earth, known as



the Mariana Trench, is located in the northwestern Pacific Ocean. The crescent-shaped trench is positioned roughly between 140ºE, 10ºN and 150ºE, 60ºN (Gvirtzman and Stern, 2004). The model eastern boundary is limited to 141ºE to avoid numerical instabilities that may arise due to steep bathymetric slopes such as the Mariana Trench. Both the atmospheric and ocean components of the coupled system are selected to have the same domain. The horizontal resolution of the MC coupled model

retained to be same (4.5 km x 4.5 km) as that of the WMCao. The MC atmosphere-ocean coupled model configuration is referred to as MCao in this manuscript. .

## 2.2 Atmospheric model

The atmospheric component of T18 has been improved to employ the SINGV v5 science configuration described in Huang et al. (2019), which is similar to the Regional Atmosphere v1 in the Tropics (RA1T) configuration of MetUM as described in

Bush et al. (2020). The model is employed operationally by the Meteorological Service of Singapore since 2019 at a higher resolution (1.5 km) for the region 95ºE to 109ºE, 6ºS to 8ºN and is referenced in the literature as SINGV. The key differences of $MC_{ao}$ atmospheric model component to T18 are;

- Lateral boundary conditions are provided at 3 hourly frequency from the deterministic ECMWF forecasts instead of the MetUM global deterministic model. This change has led to a significant increase in precipitation skill scores
across all spatial scales and precipitation thresholds in SINGV (Huang et al., 2019);

- The model uses a Prognostic Cloud fraction and Prognostic Condensate scheme (PC2, Wilson et al., 2008), instead of the diagnostic scheme of Smith (1990). This change helped to reduce the occurrence of spurious convection with very high rainfall rates and resulted in a better organization of convection, as shown in Dipankar et al. (*manuscript submitted*)

The rest of the model formulations are similar to T18 and the SINGV configuration as described in Huang et al. (2019). Main characteristics of the model are summarized below;

- The dynamical core is the non-hydrostatic semi-Lagrangian and semi-implicit Even Newer Dynamics for the General Atmospheric Modelling of the Environment (ENDGAME, Wood et al., 2014), with an Arakawa C staggered grid. The model time step is 120 s.

- A terrain-following vertical coordinate with a resolution of 80 levels and a top lid at 38.5 km. Vertical resolution is of 5 m at the boundary layer and 1.45 km below the model top, similar to the SINGV configuration.

- Boundary Layer parametrization is based on a blending between the one-dimensional scheme of Lock et al. (2001) and the three-dimensional Smagorinsky-Lilly scheme (Lilly, 1962), blending is described in Boutle et al. (2014).

- Microphysics scheme is based on Wilson and Ballard (1999) with prognostic rain formulation and improved particle
size distribution for rain as in Abel and Boutle (2012).





- Radiation scheme is based on the Edwards and Slingo (1996) scheme, with six bands in the shortwave and nine bands in the longwave (Manners et al., 2011).

- The Joint UK Land Environment Simulator (JULES, Best et al., 2011) land surface scheme with 9 surface fraction types.

- The moist conservation scheme as described in Aranami et al. (2015).

Atmospheric component of the $MC_{ao}$ employed in this study is referred to as $MCA_{ao}$ hereafter.

## 2.3 Ocean model

A regional version of Ocean PArallelise ocean engine within the NEMO (version 3.6_stable, revision 6232, Madec et al., 2016) framework is employed as the oceanic component of the $MC_{ao}$. NEMO is a primitive equation, hydrostatic, Boussinesq ocean

model extensively used in climate and operational forecast applications. The $MC_{ao}$ ocean configuration shares many features of its predecessor, $WMC_{ao}$. Hence, only key features of the NEMO and main updates of $MC_{ao}$ configuration are discussed here.

The model horizontal grid is in orthogonal curvilinear coordinates, with Arakawa C-grid staggering. The bathymetry of $MC_{ao}$ is based on the General bathymetric Chart of the Oceans (GEBCO2014) 30-arc second data. The model has 51 vertical levels

in terrain-following coordinate system and uses the stretching function by Siddorn and Furner (2013). The stretching function maintains a near-uniform surface cell thickness ($\leq$ 1 m) and hence ensures the consistent exchange of air-sea fluxes over the domain, which is critical in the atmosphere-ocean coupling. Non-linear free surface following the variable volume layer formulation by Levier et al. (2007) is used for model free surface computation. The ocean model configurations used in our study have baroclinic and barotropic time steps of 120 s and 8 s, respectively.

The Generic Length Scale (GLS) turbulence model (Umlauf and Burchard, 2013) with K-ε turbulent closure scheme and the stability function from Canuto et al. (2001) are used to compute the turbulent viscosities and diffusivities. Background vertical eddy viscosity and eddy diffusivity coefficients are set to a lower value of $1.2 \times 10^{-6}$ in $MC_{ao}$, whereas these coefficients were $1.2 \times 10^{-4}$ and $1.2 \times 10^{-5}$, respectively, in the $WMC_{ao}$. Additional vertical mixing resulting from internal tide breaking is parameterized in the model as proposed by St. Laurent et al. (2002). Both energy and enstrophy conserving scheme is used for

the momentum advection. For lateral tracer diffusion, the Laplacian operator along geopotential levels with a coefficient of 20 $m^2 s^{-1}$ is used, while iso-level bilaplacian viscosity with a coefficient of $-6 \times 10^7$ $m^2 s^{-1}$ is applied for the momentum mixing. Implicit form of non-linear parameterization with a log-layer formulation is used for the bottom drag coefficient computation. The minimum and maximum of the drag coefficient are set to 0.0001 and 0.15, respectively.

At the lateral open ocean boundaries, the flow relaxation scheme (FRS, Davies, 1976) is applied for the tracers and baroclinic

velocities, while Flather boundary condition (Flather, 1976) is used for the sea surface height (SSH) and barotropic velocities.



One of the key updates to MC$_{ao}$ is the implementation of tide forcing at the lateral boundaries and tide potential at the ocean surface. Due to certain numerical issues, the tide related forcings are not included in the WMC$_{ao}$. The tidal elevations and currents from Finite Element Solutions (FES2014b) data have been used for providing the tidal harmonics at the lateral boundaries. Fifteen major tidal constituents (Q1, O1, P1, S1, K1, 2N2, Mu2, Nu2, N2, M2, L2, T2, S2, K2 and M4) are

included in the boundary forcing.

Both coupled and uncoupled ocean model configurations are employed in the study. For uncoupled simulations, the air-sea heat fluxes are estimated using the Common Ocean-ice Reference Experiment (CORE) bulk formulae (Large and Yeager, 2004). However, a direct flux formulation is used in the coupled ocean model. Monthly runoff climatology from Dai and Trenberth (2002) and chlorophyll monthly climatology from SeaWiFS satellite observation are provided as runoff forcing and

to compute light absorption coefficients, respectively, in all ocean configurations. The Red-Blue-Green (RGB) scheme is used to calculate the penetration of shortwave radiation into the ocean (Lengaigne et al., 2007). Identical to WMC$_{ao}$, the fraction of solar radiation absorbed at the surface layer is defined to be 56% of the downward component. Mean sea level pressure (MSLP) forcing is included in the surface boundary forcing to take account of the inverse barometric effect on SSH.

The uncoupled and coupled ocean model configurations employed in this study are referred to as MCO and MCO$_{ao}$,

respectively.

### 2.4 Coupled configuration

The exchange of fluxes between the atmosphere and ocean models is achieved through the Ocean Atmosphere Sea ice Soil coupler (version 3.3) interfaced with the Model Coupling Toolkit (OASIS3-MCT) libraries (Valcke, 2013). The Earth System Modelling Framework (ESMF) regrid tools are used to generate the interpolation weights for the remapping of exchange fields.

The coupling occurs at hourly frequency and hourly mean fields are exchanged. Since a direct flux formulation is implemented, the heat fluxes computed using the Monin-Obukhov similarity theory is exchanged from the atmosphere to the ocean model. The sea surface temperature (SST) and zonal and meridional surface current fields are sent from the ocean to the atmosphere model. The variables exchanged from atmosphere to the ocean include; non-solar heat flux, net shortwave radiation, liquid precipitation, net evaporation, and zonal and meridional wind stress. Due to numerical issues, MSLP exchange from the

atmosphere is not enabled in the MC$_{ao}$. Instead, it is supplied from an external data source to the ocean model.

### 2.5 Model initialization and forcing

To assess the performance of the ocean model and provide initial condition to the MCO$_{ao}$, a 69-month hindcast simulation is performed with MCO for the period 01 January 2014 to 30 September 2019. The MCO is initialized in 1 January 2014 using temperature, salinity, zonal and meridional currents, and SSH derived from Mercator global ocean reanalysis. The lateral

boundary condition for the hindcast simulation is also obtained from the same ocean reanalysis data. The daily mean of temperature, salinity, baroclinic and barotropic velocities, and SSH are included in the lateral boundary forcing. Ocean surface





is forced by ECMWF Reanalysis 5 (ERA5) during the period from 1 January 2014 to 30 June 2019. Downward shortwave and longwave radiation at the ocean surface, total precipitation, MSLP, and 10-m wind velocities, air temperature and specific humidity fields are included in the forcing file. Since there was a delay of about 2-3 months in the release of ERA5 data during

the time of model development, the MCO is forced by the 6-hourly ECMWF IFS analysis fields from 1 July 2019 to the start of MC$_{ao}$ pre-operational forecast run on 1 October 2019 (Figure 2a). As the atmospheric adjustments are sub-daily, no spin-up/hindcast simulations are performed for the MCA$_{ao}$.

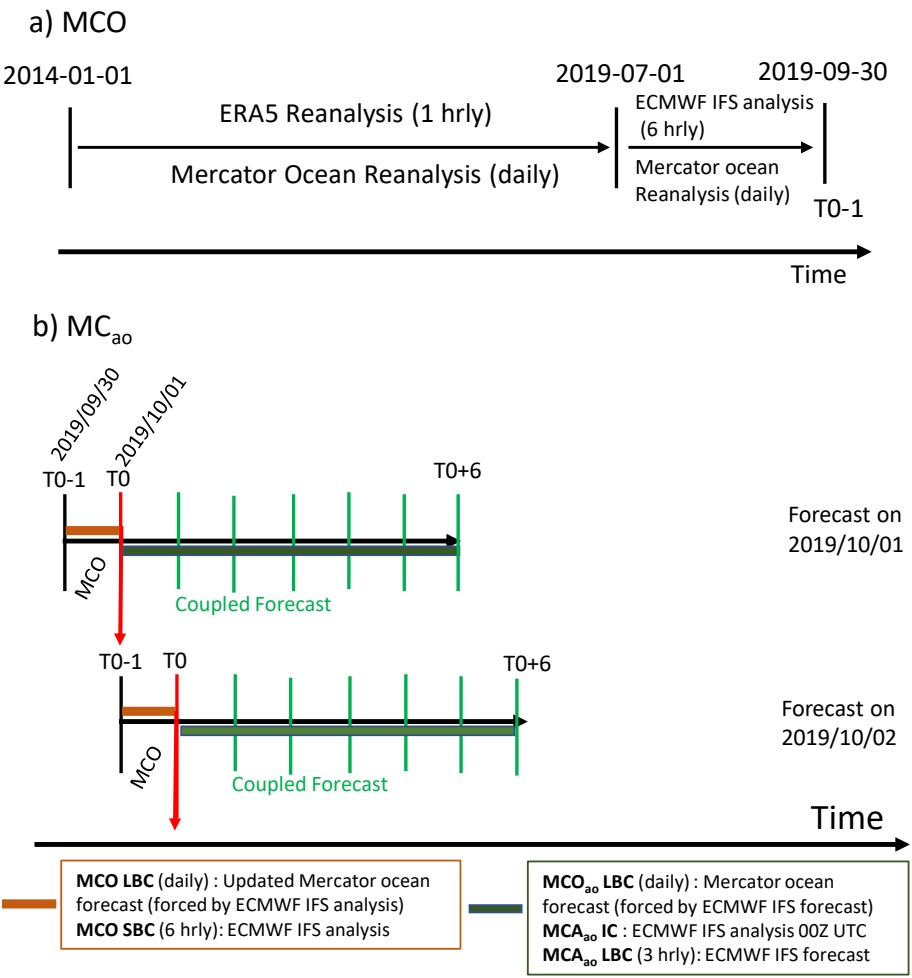

**Figure 2:** Schematic of modelling systems used in the study (a) MC ocean only model (MCO) hindcast and (b) MC atmosphere-ocean coupled forecast model (MC$_{ao}$). MCO$_{ao}$ – MC coupled Ocean model, MCA$_{ao}$ – MC coupled Atmospheric model, LBC – Lateral Boundary Condition, SBC – Surface Boundary Condition, IC – Initial Condition.





A schematic of the atmosphere-ocean coupled system used in the pre-operational forecast is shown in Figure 2b. In the coupled
prediction system, the MCA$_{ao}$ is initialized daily at 00Z UTC from the ECMWF IFS analysis. MCO run for the previous day
(T0 minus 1), forced by 6-hourly ECMWF IFS analysis at the surface and the daily mean of updated Mercator ocean forecast
as the LBC, provides the initial condition to MCO$_{ao}$. Since it is driven by analyzed (or updated) surface (lateral) boundary
conditions, the MCO provides an updated initial condition to the MCO$_{ao}$ daily. The MC$_{ao}$ forecast run is driven by LBC from
3-hourly ECMWF IFS forecasts in the atmosphere and daily Mercator forecasts in the ocean. Since the MSLP from MCA$_{ao}$ is
not incorporated in MCO$_{ao}$, 3-hourly EMCWF IFS forecast data is supplied to the model.

**2.6 Pre-operational forecast setup**

The atmosphere-ocean coupled forecast model has run as a pre-operational forecast system from 1 October 2019 to 31 October
2019 at the Cray XC-40 HPC located in the Center for Climate Research Singapore (CCRS), Singapore. The forecast system
includes all necessary programs/scripts for the pre-processing of atmospheric and oceanic variables to their respective model
grids. The forecast system is scheduled to initialize the forecasts daily at 1300 UTC and simulations are completed by ~1840
UTC. Summary of HPC resources usage and typical runtimes for daily forecast simulations are shown in Table 1. To minimize
the output size, only basic oceanic and atmospheric variables are included in the output. The forecast from MCO$_{ao}$ includes
instantaneous SSH, hourly averaged sea surface temperature, sea surface salinity, and surface current velocities, and daily
mean of ocean temperature, salinity and ocean currents. Further, to test the feasibility of the coupled forecast system for
operational purpose, we have conducted simulations with increased computational resources. Test simulations showed that by
increasing the computational resources to 81 nodes (2916 cores), the total runtime has been reduced to ~140 min. This suggests
a near-linear reduction in total runtime with an increase in computation nodes.

| Configuration | Uncoupled ocean (MCO) | Coupled Atmosphere (MCA$_{ao}$) | Coupled Ocean (MCO$_{ao}$) |
|---|---|---|---|
| Total nodes (cores) | 16 (576) | 24 (864) | 4 (144) |
| Daily runtime | 6.5 min | 330 min | |
| Core hours | 3.9 | 198 | |
| Flume/IO (node) | - | 1 | |

**Table 1:** Summary of HPC resources usage and typical runtimes.



## 3 Data


A brief description of the reanalysis, forecast and observational datasets used for the model initialization, forcing and evaluation is presented in this section.

### 3.1 Model initialization and forcing

ERA5 is a climate reanalysis produced by the ECMWF providing hourly estimates of many atmospheric, land and oceanic
fields (Hersbach et al., 2020). Currently, it covers the period from 1979 to within 5-days of present-time and horizontal resolution is approx. 30 km. The reanalysis produced using 4D-Var assimilation of ECMWF Integrated Forecast System (IFS). ERA5 combines vast amounts of historical observations into global estimates using advanced modelling and data assimilation systems. The data is freely available through the data server https://cds.climate.copernicus.eu/.

ECMWF IFS is a global weather prediction system comprising a spectral atmospheric model, ocean wave model, ocean model
and land surface model coupled to a 4D-Var data assimilation system. IFS medium-range weather forecasts are available up to 10 days at a horizontal resolution of 0.1°. In addition, the atmospheric analysis fields are provided four times daily for the forecast base time 00, 06, 12 and 18 UTC. The data is available to registered users from https://www.ecmwf.int/en/forecasts/datasets/

Mercator global ocean reanalysis/forecast provides oceanic variables with 1/12° horizontal resolution. The system uses NEMO
v3.1 with 50 vertical z-levels ranging from zero to 5500 m and forced by the ECMWF IFS meteorological variables. The assimilation/forecast product includes the daily mean of temperature, salinity, currents from top to bottom over the global ocean and SSH. The data is freely available from https://marine.copernicus.eu/.

The tidal heights and currents computed from the global tide model Finite Element Solution (FES2014b) is used as the tidal forcing in the model. FES2014 is based on the resolution of the shallow water hydrodynamic equations (T-UGO model) in a
spectral configuration and using a global finite element mesh with increasing resolution in coastal and shallow waters regions (Lyard et al., 2006). The database is distributed on a global 1/16° x 1/16° grid. Data is produced by assimilating long-term altimetry data (Topex/ Poseidon, Jason-1, Jason-2, TPN-J1N, and ERS-1, ERS-2, ENVISAT) and tidal gauges through an improved representer assimilation method. Tidal heights and currents of 32 tidal constituents are available. The data is freely available through http://www.aviso.altimetry.fr/en/data/products/auxiliary-products/global-tide-fes.html.

### 3.2 Model evaluation


The CORIOLIS data service provides quality-controlled is situ data in real-time and delayed mode over the global ocean. The data include temperature and salinity profiles and time series from profiling floats, Expendable Bathy Thermograph's (XBT), thermo-salinographs (TSG), and drifting buoys. The data is freely available from http://www.coriolis.eu.org/Data-Products/.

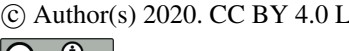


The Operational Sea Surface Temperature and Sea Ice Analysis (OSTIA) SST is produced daily on an operational basis at the
UK Met Office using optimal interpolation on a global 0.054º x 0.054º grid. The product assimilates satellite data including
advanced very-high resolution radiometer, Spinning Enhanced Visible and Infrared imager, Geostationary Operational
Environmental Satellite Imager, Infrared Atmospheric Sounding Interferometer, Tropical Rainfall  Measuring Mission
Microwave imager and in situ data from ships, drifting and moored buoys (Donlon et al., 2012). SST data at every grid point
is accompanied by an uncertainty estimate, known as an analysis error, and an optimal interpolation approach is employed to
produce this estimate. The data is freely available from https://marine.copernicus.eu/.

The University of Hawaii Sea Level Center (UHSLC) offers quality controlled tide gauge (TG) sea level observations over the
global ocean as fast delivery (FD, 1-2 months delay) and research quality (RQ, 1-2 year delay) data at hourly and daily
resolution (Caldwell et al., 2015). The data is freely available from http://uhslc.soest.hawaii.edu/data/.

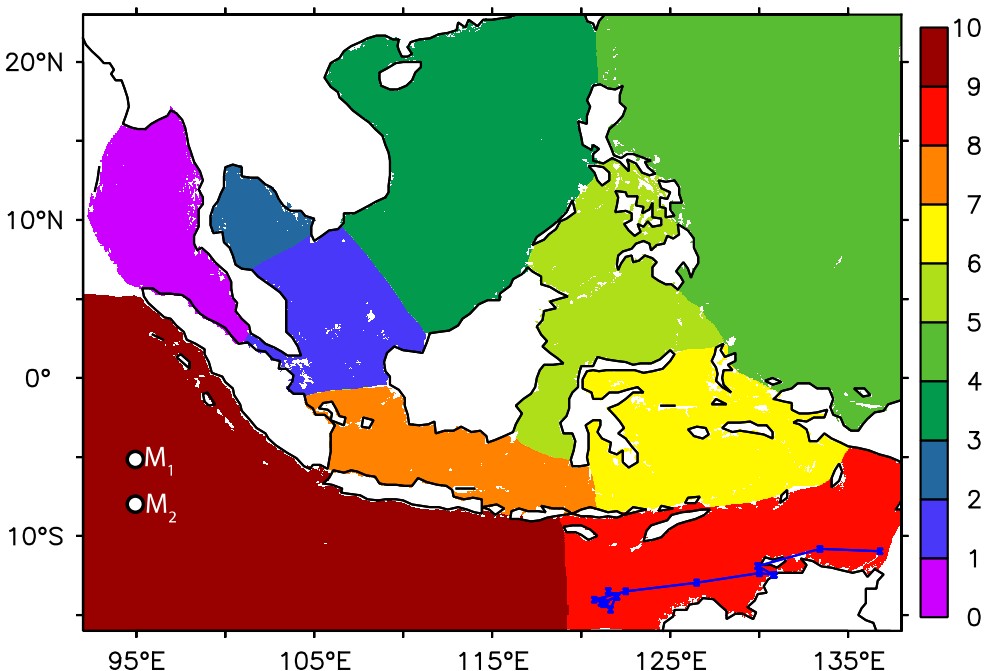

**Figure 3:** Domain used for hindcast/forecast evaluation with sub-regions defined in the study. 1. Andaman Sea-Malacca
Strait (ASMS), 2. Southern South China Sea (SSCS), 3. Gulf of Thailand (GoT), 4.  Rest of the South China Sea (RSCS),
5. Tropical Western Pacific Ocean (TWPO), 6. Sulu-Celebs Sea (SuCeS), 7. Banda Sea (BS), 8. Java Sea (JS), 9. Timor-
Arafura Sea (TAS) and 10. Tropical eastern Indian Ocean (TEIO). The Bay of Bengal region (north of $5^0$N, west of $92^0$E)
is excluded from the analysis-domain. Blue line in the TAS indicates the TSG observation track. Moored buoy locations
$M_1$ ($95^0$E, $5^0$S) and $M_2$ ($95^0$E, $8^0$S) are also shown.





## 4 Results and discussion


An evaluation of the MCO hindcast and MCO$_{ao}$ forecast simulations are presented in this section of the manuscript. Direct comparison of model simulations with observation or analysis data has been performed. Based on the availability of in situ or satellite observation at the time of data analysis, only a few variables are selected for assessing the model performance. Also, to maintain consistency between the evaluation of hindcast and forecast simulations, analyses of the same set of variables and

observation data have been performed where possible. Oceanic variables employed for the evaluation are SST, SSH, and the subsurface temperature and salinity.

Model hindcast/forecast over the region 92ºE to 138ºE and 16ºS to 23ºN, defined as analysis-domain (Figure 1 and 3), is further used for the analysis. The MC model domain includes oceanic basins with different geographical and climatological characteristics. For the evaluation purpose, we have divided the analysis-domain into 10 sub-regions based on their

geographical distribution (Figure 3). These sub-regions are; 1. Andaman Sea-Malacca Strait (ASMS), 2. Southern South China Sea (SSCS), 3. Gulf of Thailand (GoT), 4. Rest of the South China Sea (RSCS), 5. Tropical Western Pacific Ocean (TWPO), 6. Sulu-Celebs Sea (SuCeS), 7. Banda Sea (BS), 8. Java Sea (JS), 9. Timor-Arafura Sea (TAS) and 10. Tropical eastern Indian Ocean (TEIO).

### 4.1 Ocean Hindcast

An overall assessment of the ocean model hindcast simulation is carried out to understand the realism of the ocean initial condition for the coupled forecasts, particularly at the ocean surface where the exchange of fluxes between the atmosphere and ocean takes place. Though the hindcast simulations are encompassed from 01 January 2014 to 30 September 2019, we only evaluate ERA5 driven simulations during the period from 1 January 2018 to 30 June 2019. First 4 years of simulation is considered as the spin-up stage of the model. Comparison of daily mean SST with OSTIA analysis and moored buoys

observation is presented while the daily mean SSH is compared with tide-gauge observations. Moored observation buoys in the eastern tropical Indian Ocean established as part of the Research Moored Array for African-Asian-Australian Monsoon Analysis (RAMA, McPhaden et al., 2009) at the locations 95ºE, 5ºS (M$_1$) and 95ºE, 8ºS (M$_2$) are used for the evaluation. Fast delivery (FD) data from UHSLC for 20 tide gauge stations are employed for the SSH comparison.

### 4.1.1 Sea surface temperature

Comparison of model simulated daily mean SST with OSTIA analysis is shown in Figure 4. Spatial distribution of model SST bias (Figure 4a), root mean square difference (RMSD) (Figure 4b), correlation coefficient (Figure 4c) and the spatial average of SST difference over the analysis-domain (Figure 4d) are given. Model performance in simulating SST over the sub-regions is given in Table 2. SST Bias, RMSD and correlation coefficient statistics computed using modelled SST and OSTIA are





shown in the Table. The SST bias is within ±0.2 °C for about 76% and within ±0.5 °C for about 98% of the analysis domain.
Largest SST cold bias is seen in the Andaman Sea region. Meanwhile, most of the South China Sea (SCS), equatorial west
Pacific Ocean and Australian coasts of the Timor Sea show a positive SST bias. Negative SST bias of about -0.25 °C is
observed in the ASMS region, while positive bias over 0.25 °C is confined to the SSCS and GoT sub-regions. Rather than
appearing as a basin-wide feature, higher positive biases appear as small circular patches in the northern SCS region that
represents the likely existence of cyclonic eddies over this region.

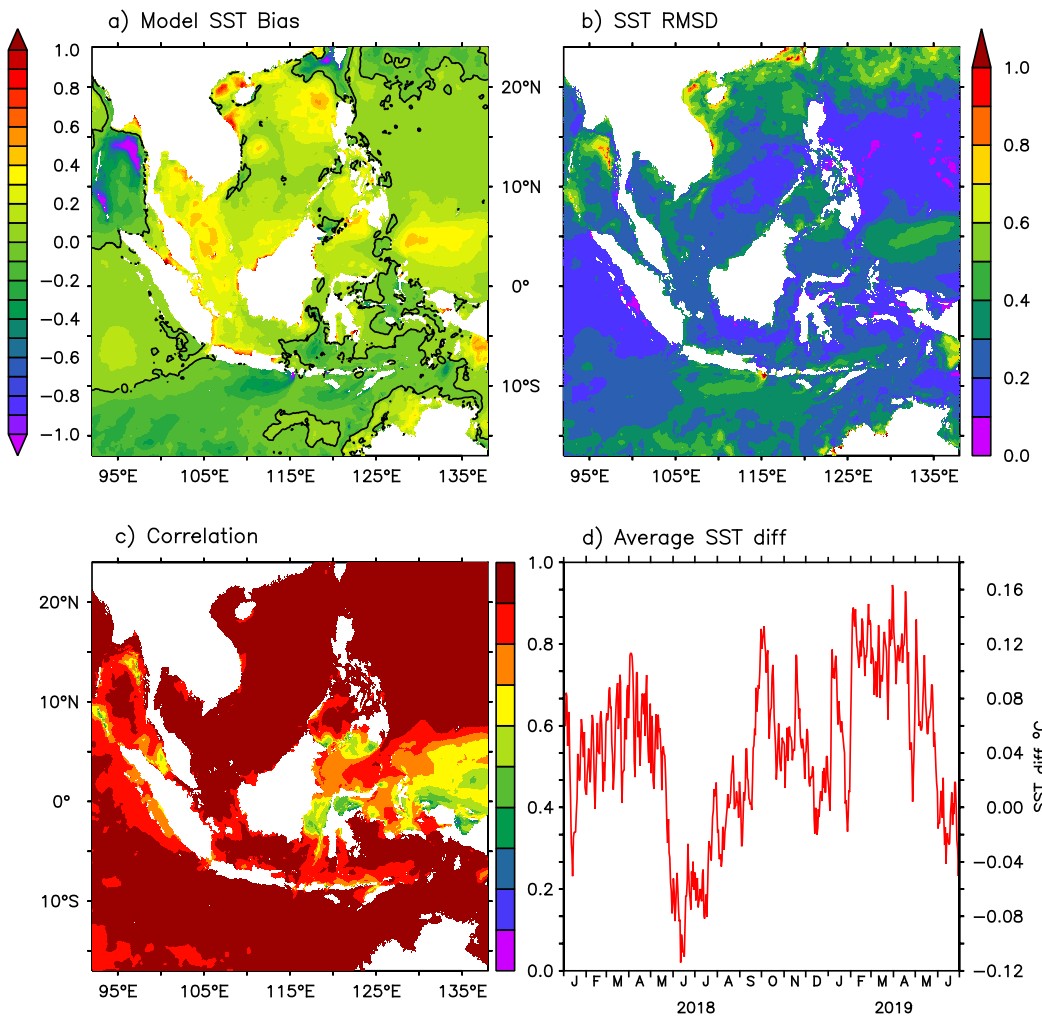

**Figure 4:** Spatial distribution daily averaged (a) SST bias (°C), (b) RMSD (°C) and (c) correlation coefficient between
the MCO hindcast and OSTIA analysis. (d) Spatial average of SST difference between model and OSTIA over the
analysis-domain (°C).






| No | Region | Bias (ºC) | RMSD (ºC) | Correlation Coefficient |
|---|---|---|---|---|
| 1 | Andaman Sea & Malacca Strait | -0.25 | 0.53 | 0.84 |
| 2 | Southern SCS | 0.29 | 0.30 | 0.95 |
| 3 | Gulf of Thailand | 0.26 | 0.30 | 0.94 |
| 4 | Rest of SCS | 0.16 | 0.39 | 0.96 |
| 5 | Tropical West Pacific Ocean | 0.07 | 0.29 | 0.88 |
| 6 | Sulu-Celebes Sea | 0.12 | 0.31 | 0.80 |
| 7 | Banda Sea | 0.00 | 0.25 | 0.84 |
| 8 | Java Sea | 0.09 | 0.30 | 0.89 |
| 9 | Timor-Arafura Sea | 0.01 | 0.34 | 0.95 |
| 10 | Tropical East Indian Ocean | -0.05 | 0.30 | 0.92 |
| **Mean Bias, RMSD and mean correlation** | | **0.07** | **0.34** | **0.90** |

**Table 2:** Summary of SST bias, RMSD and correlation coefficient statistics between model hindcast and OSTIA for the period 01/01/2018 to 30/06/2019. Daily mean SST from model and OSTIA is used for the analysis.

The RMSD between model and OSTIA is less than 0.5 ºC for about 97% of the analysis-domain (Figure 4b). Small patches of higher RMSD (> 0.7 ºC) are mostly seen along the coastal regions. The RMSD minimum (0.25 ºC) and maximum (0.53 ºC)

are observed over the BS and ASMS sub-regions, respectively (Table 2). Correlation between the model SST hindcast and OSTIA is above 99.9% confidence level over the analysis-domain (Figure 4c). Over 88% of the domain displays a correlation higher than 0.8. Relatively low correlation is seen over the middle of Malacca Strait, Makassar Strait and equatorial Pacific Ocean regions. In sub-region spatial average, the lowest (0.8) and highest (0.96) correlations are seen over the SuCeS and RSCS regions, respectively (Table 2). Time series of spatially averaged SST difference between model and OSTIA is shown

in Figure 4d. Consistent with our earlier analyses, relatively low SST difference depicts a good agreement between the MCO SST hindcast and OSTIA analysis. Further analysis of SST in different sub-regions revealed that relatively higher SST over the GoT, SSCS and SuCeS regions contribute to the positive SST differences during February-April in 2018-2019 and September-October in 2018 (figures not shown). Meanwhile, SST simulation over those sub-regions shows improvement during June-July 2018. Higher negative SST bias over the ASMS region mainly contributes to the negative SST difference

during the same period. Overall, the mean SST bias, RMSD and mean correlation over the analysis-domain are 0.07 ºC, 0.34 ºC and 0.90, respectively.

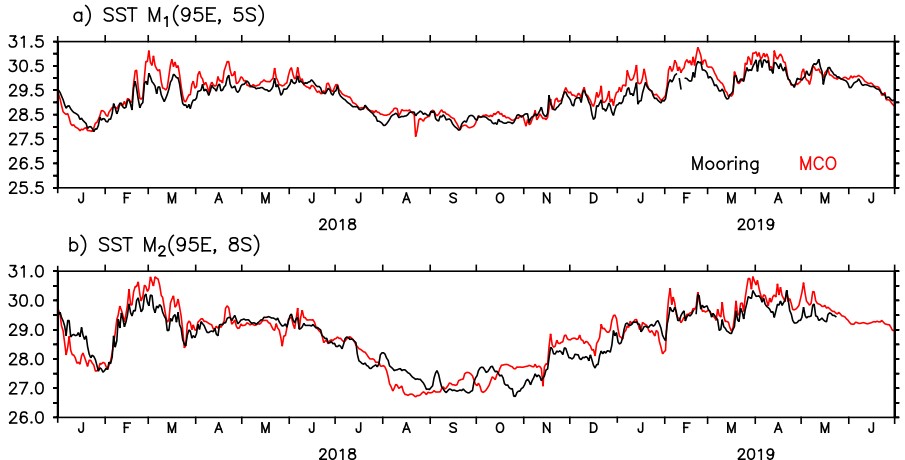

**Figure 5:** Comparison of daily averaged SST from MCO hindcast and RAMA moored buoys at (a) $M_1$ and (b) $M_2$ locations during the period 1 January 2018 to 30 June 2019.

Time series of daily mean SST from the RAMA moored observation buoys located in the southeastern tropical Indian Ocean at 95ºE, 5ºS ($M_1$) and 95ºE, 8ºS ($M_2$) is shown in Figure 5. Model SST is bilinearly interpolated to the buoy locations. Temperature observation at 1 m depth is taken as SST from the moored buoys while temperature averaged over upper 1 m is

indicated as the model SST at these locations. In general, a good agreement is found between the model and observations at both mooring locations. Both the seasonal and intra-seasonal SST variability is reasonably well reproduced by the model. SST bias, RMSD and correlation between the model and observation are 0.17 ºC, 0.29 ºC and 0.94, respectively, for $M_1$ and 0.12 ºC, 0.41 ºC and 0.92, respectively, for $M_2$ locations. The standard deviation (SD) of SST at $M_1$ and $M_2$ locations are 0.94 ºC and 0.99 ºC, respectively, and the RMSD is relatively smaller than the SD at both locations.

**4.1.2 Sea surface height**

Daily mean SSH observation from 20 tide-gauge stations distributed across the domain and MCO simulated SSH interpolated to the location of these observations have been used for the hindcast evaluation. SSH bias, RMSD and correlation coefficient statistics between model and SSH observations are given in Table 3. Highest SSH bias (0.12 m) and RMSD (0.15 m) are seen at the Malakal, Palau, tide-gauge station. The SSH bias is within ±0.05 m for 17 of the total 20 stations analyzed. The model

accuracy is higher than 0.10 m for 18 stations while 14 of the total 20 stations have an accuracy greater than 0.05 m. The SSH correlation between model and observation is above 99.9% confidence level for all tide-gauge stations employed in the analysis. The correlation is above 0.80 for 16 tide-gauge stations. Lowest correlation of 0.60 is observed at the Malakal tide-





gauge station. Mean SSH bias, RMSD and mean correlation between the model and observation are 0.01 m, 0.06 m and 0.87, respectively.

| No | Station name & Country | Latitude, Longitude | Bias (m) | RMSD (m) | Correlation Coefficient |
|---|---|---|---|---|---|
| 1 | Sabang, IDN | 5.888N, 95.317E | 0.00 | 0.04 | 0.85 |
| 2 | Sibolga, IDN | 1.75N, 98.767E | 0.01 | 0.03 | 0.90 |
| 3 | Padang, IDN | 1.0S, 100.367E | -0.05 | 0.06 | 0.89 |
| 4 | Cilicap, IDN | 7.752S, 109.017E | 0.00 | 0.04 | 0.93 |
| 5 | Prigi, IDN | 8.28S, 111.73E | 0.00 | 0.05 | 0.96 |
| 6 | Benoa, IDN | 8.745S, 115.21E | 0.00 | 0.04 | 0.94 |
| 7 | Saumlaki, IDN | 7.982S, 131.29E | 0.02 | 0.05 | 0.81 |
| 8 | Bitung, IDN | 1.44N, 125.193E | 0.06 | 0.07 | 0.73 |
| 9 | Malakal, PLW | 7.33N, 134.463E | 0.12 | 0.15 | 0.60 |
| 10 | Davao Gulf, PHL | 7.122N, 125.663E | 0.00 | 0.03 | 0.88 |
| 11 | Subic Bay, PHL | 14.765N, 120.252E | 0.04 | 0.05 | 0.94 |
| 12 | Manila, PHL | 14.585N, 120.968E | -0.01 | 0.04 | 0.95 |
| 13 | Legaspi, PHL | 13.15N, 123.75E | 0.00 | 0.03 | 0.83 |
| 14 | Currimao Ilocos Norte, PHL | 17.988N, 120.488E | 0.00 | 0.05 | 0.96 |
| 15 | Hong Kong, HKG | 22.3N, 114.2E | -0.02 | 0.09 | 0.75 |
| 16 | Qui Nhon, VNM | 13.775N, 109.255E | 0.01 | 0.05 | 0.90 |
| 17 | Vung Tau, VNM | 10.34N, 107.072E | 0.01 | 0.07 | 0.90 |
| 18 | Ko Lak, THA | 11.795N, 99.817E | -0.01 | 0.06 | 0.95 |
| 19 | Ko Taphao Noi, THA | 7.832N, 98.425E | 0.00 | 0.04 | 0.91 |
| 20 | Pulau Langkawi, MYS | 6.432N, 99.765E | -0.03 | 0.14 | 0.72 |
| **Mean Bias, RMSD and mean Correlation** | | | 0.01 | 0.06 | 0.87 |

**Table 3:** Summary of SSH bias, RMSD and correlation coefficient statistics between MCO hindcast and tide-gauge stations for the period 01/01/2018 to 30/06/2019. Daily mean SSH from model and tide-gauge is used for the analysis.

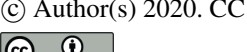



Time series of daily mean SSH from model and observations for randomly selected stations are plotted in Figure 6. Where the stations Sibolga and Prigi are located in the eastern tropical Indian Ocean, and Currimao Ilocos Norte and Vung Tau are located in the SCS. Generally, the model simulated SSH follows the observation and shows good agreement with it. A few sharp peaks

in the tide-gauge observation are found to be absent in the model simulation (e.g. Figure 6c). Most of the tide gauge stations are located adjacent to the coast and our current model resolution is not enough to resolve the coastline at very fine scales. Since the model SSH is interpolated to the tide gauge location, local scale SSH variations may not be captured in the model simulation. This may be one possible reason for the discrepancy between model and observation.

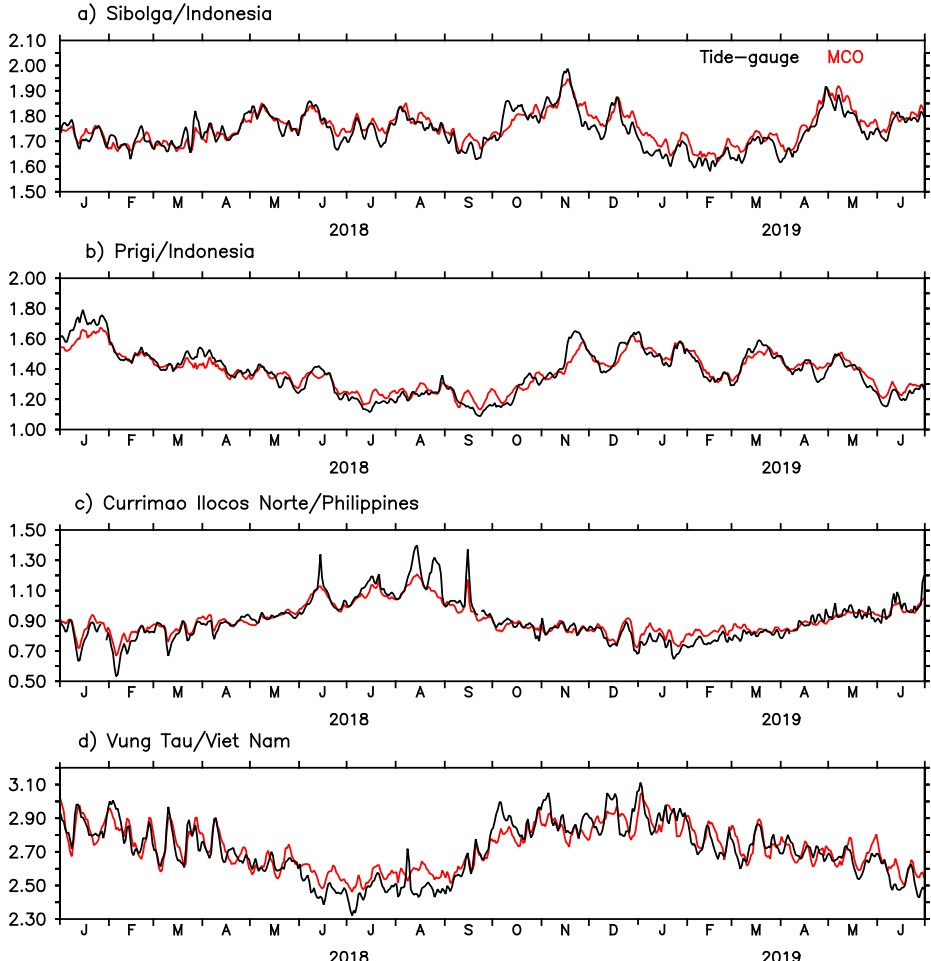

**Figure 6:** Time series of daily mean SSH (in m) from tide-gauge observation (black line) and MCO hindcast (red line) at randomly selected stations (a) Sibolga (1.75$^0$N, 98.76$^0$E), (b) Prigi (8.28$^0$S, 111.73$^0$N), (c) Currimao Ilocos Norte (17.988$^0$N, 120.488$^0$E) and (d) Vung Tau (10.34$^0$N, 107.072$^0$E) during 1 January 2018 to 30 June 2019.





Comparison of model simulated SST and SSH fields show good agreement with observation and analysis data. The RMSD and bias statistics of SST and SSH relative to the observation are within the acceptable error limits of ocean hindcast simulations (e.g. Yang et al., 2016). Statistically significant correlation with observation suggests that both the spatial and temporal patterns of variability are reasonably well reproduced by the model.

## 4.2 Ocean Forecasts

Results from the analysis of MCO$_{ao}$ forecast simulations for October 2020 are presented here. Since the system delivers a 6-day forecast, the analysis period extends from 1 October to 5 November 2020. Daily files are produced at different forecast lead times, T+0 to T+24 (fsct_day1), T+24 to T+48 (fcst_day2), T+48 to T+72 (fcst_day3), T+72 to T+96 (fcst_day4), T+96 to T+120 (fcst_day5) and T+120 to T+144 (fcst_day6) for the following analyses. Comparisons of coupled ocean forecast for different forecast lead times with OSTIA SST and in situ observations such as temperature from RAMA moored buoys, TSG

and XBT profiles, temperature and salinity from Conductivity Temperature Depth (CTD) profiles, and SSH from tide-gauges have been performed.

### 4.2.1 Sea surface temperature

Time series of daily mean SST averaged over the sub-regions from OSTIA analysis and different forecast lead times are plotted in Figure 7. Statistics of SST bias, RMSD and correlation coefficient between model and OSTIA for the October forecast run

are listed in Table 4. The forecasted SST over most of the sub-regions is within the error standard deviation of the OSTIA analysis, which is indicated by shading in Figure 7. Excluding the ASMS, all other sub-regions exhibit a warm SST bias with the largest values over the SSCS and GoT. The RMSD is less than 0.5 ºC over most of the sub-regions during the analysis period. Over ASMS sub-region, both the cold bias and RMSD increases with the forecast lead time and shows the highest RMSD (0.49 ºC) on fsct_day6. The largest SST RMSD (0.56 ºC) and bias (0.49 ºC) over the entire sub-regions is observed

over the GoT on 1-day forecast lead time. Generally, the forecasted SST tends to be cooler with an increase in forecast lead time, denoting a lower warm bias and RMSD relative to fcst_day1. Interestingly, inconsistent with the improvements in SST bias and RMSD, the correlation between forecasted and OSTIA time series considerably decreases with higher forecast lead times (Table 4).

SST correlation is above 95% confidence level (r > 0.365) over entire analysis-domain during fcst_day1. In general, SST

correlation is higher than 99% confidence level (r > 0.46) over 60% of the sub-regions during higher forecast lead times as well. The sub-regions including ASMS, GoT, TWPO and SuCeS are noted by lower correlation significance level with an increase in forecast lead time. Overall, the SST correlation over the analysis-domain is above 99% confidence level across all forecast lead times while bias and RMSD are less than 0.19 ºC and 0.35 ºC, respectively.



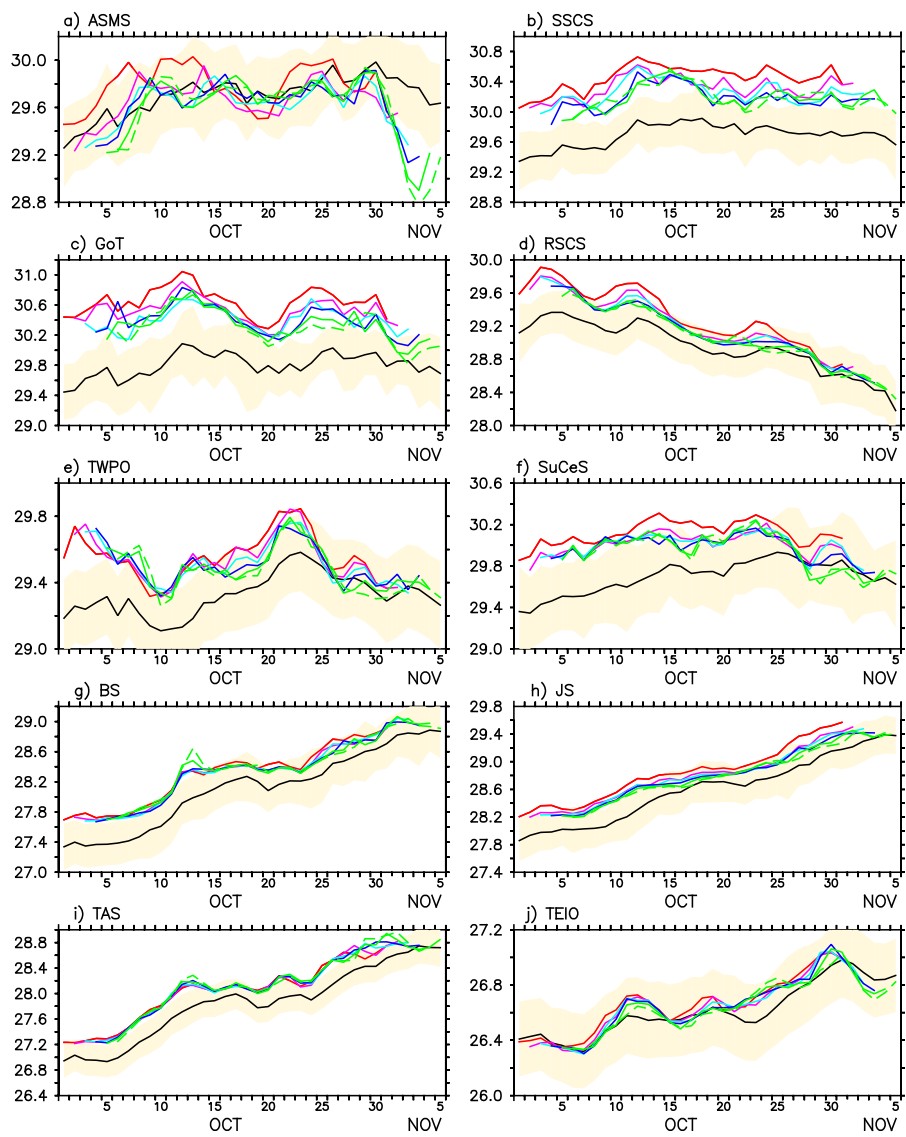

**Figure 7:** Time series of daily mean SST from model forecast and OSTIA averaged over the sub-regions. OSTIA (black line), fcst_day1 (red line), fcst_day2 (purple line), fcst_day3 (light blue line), fcst_day4 (blue line), fcst_day5 (green line) and fcst_day6 (green dash). Shading represents the estimated error standard deviation of OSTIA analyzed SST. (a) ASMS: Andaman Sea-Malacca Strait, (b) SSCS: Southern SCS, (c) GoT: Gulf of Thailand, (d) RSCS: Rest of SCS, (e) TWPO: Tropical Western Pacific Ocean, (f) SuCeS: Sulu Celebes Sea, (g) BS: Banda Sea, JS: (h) Java Sea, (i) TAS: Timor-Arafura Sea, (j) TEIO: Tropical Eastern Indian Ocean. *y*-axes are different in the plots.






(a)

| No | | Bias (${}^{0}$C) | | | | | | RMSD (${}^{0}$C) | | | | | |
|---|---|---|---|---|---|---|---|---|---|---|---|---|---|
| | | Forecast lead time (days) | | | | | | Forecast lead time (days) | | | | | |
| | | 1 | 2 | 3 | 4 | 5 | 6 | 1 | 2 | 3 | 4 | 5 | 6 |
| 1 | Andaman Sea & Malacca Strait | -0.01 | -0.11 | -0.13 | -0.14 | -0.15 | -0.17 | 0.40 | 0.41 | 0.41 | 0.43 | 0.46 | 0.49 |
| 2 | Southern SCS | 0.43 | 0.32 | 0.27 | 0.20 | 0.20 | 0.20 | 0.50 | 0.41 | 0.36 | 0.30 | 0.30 | 0.30 |
| 3 | Gulf of Thailand | 0.49 | 0.38 | 0.29 | 0.28 | 0.25 | 0.23 | 0.56 | 0.47 | 0.37 | 0.38 | 0.34 | 0.33 |
| 4 | Rest of SCS | 0.16 | 0.10 | 0.08 | 0.07 | 0.05 | 0.05 | 0.30 | 0.24 | 0.22 | 0.21 | 0.20 | 0.21 |
| 5 | Tropical West Pacific Ocean | 0.11 | 0.11 | 0.10 | 0.09 | 0.08 | 0.07 | 0.21 | 0.22 | 0.21 | 0.20 | 0.20 | 0.21 |
| 6 | Sulu-Celebes Sea | 0.21 | 0.14 | 0.13 | 0.11 | 0.10 | 0.10 | 0.33 | 0.27 | 0.27 | 0.26 | 0.26 | 0.26 |
| 7 | Banda Sea | 0.13 | 0.10 | 0.10 | 0.10 | 0.10 | 0.11 | 0.21 | 0.19 | 0.19 | 0.19 | 0.21 | 0.23 |
| 8 | Java Sea | 0.15 | 0.10 | 0.09 | 0.07 | 0.06 | 0.05 | 0.25 | 0.21 | 0.21 | 0.20 | 0.19 | 0.19 |
| 9 | Timor-Arafura Sea | 0.18 | 0.17 | 0.17 | 0.17 | 0.17 | 0.17 | 0.34 | 0.34 | 0.34 | 0.34 | 0.35 | 0.37 |
| 10 | Tropical East Indian Ocean | 0.02 | 0.02 | 0.02 | 0.02 | 0.01 | 0.01 | 0.25 | 0.23 | 0.22 | 0.22 | 0.22 | 0.22 |
| **Mean Bias and RMSD** | | **0.19** | **0.13** | **0.11** | **0.10** | **0.09** | **0.08** | **0.35** | **0.31** | **0.29** | **0.28** | **0.29** | **0.29** |
| | | **0.12** | | | | | | **0.30** | | | | | |



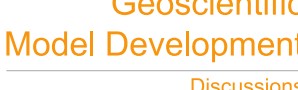

(b)

| No | | Correlation Coefficient | | | | | |
|---|---|---|---|---|---|---|---|
| | | Forecast lead time (days) | | | | | |
| | | 1 | 2 | 3 | 4 | 5 | 6 |
| 1 | Andaman Sea & Malacca Strait | 0.37 | 0.37 | 0.31 | 0.23 | 0.16 | 0.15 |
| 2 | Southern SCS | 0.63 | 0.50 | 0.52 | 0.53 | 0.51 | 0.49 |
| 3 | Gulf of Thailand | 0.48 | 0.41 | 0.38 | 0.33 | 0.30 | 0.37 |
| 4 | Rest of SCS | 0.65 | 0.63 | 0.62 | 0.63 | 0.62 | 0.61 |
| 5 | Tropical west Pacific Ocean | 0.58 | 0.49 | 0.45 | 0.39 | 0.34 | 0.30 |
| 6 | Sulu-Celebes Sea | 0.40 | 0.35 | 0.30 | 0.27 | 0.25 | 0.24 |
| 7 | Banda Sea | 0.84 | 0.82 | 0.83 | 0.80 | 0.77 | 0.72 |
| 8 | Java Sea | 0.88 | 0.88 | 0.87 | 0.87 | 0.86 | 0.85 |
| 9 | Timor-Arafura Sea | 0.83 | 0.84 | 0.84 | 0.82 | 0.79 | 0.76 |
| 10 | Tropical East Indian Ocean | 0.53 | 0.54 | 0.53 | 0.52 | 0.50 | 0.47 |
| **Mean Correlation** | | **0.62** | **0.58** | **0.57** | **0.54** | **0.51** | **0.50** |
| | | **0.55** | | | | | |

**Table 4:** Summary of SST bias and RMSD (a) and correlation coefficient (b) statistics between coupled ocean forecasts and OSTIA over the sub-regions shown in Figure 3 during 1 to 31 October 2019. Daily mean SST from model and OSTIA is used for the analysis.

Figure 8 shows the time series of hourly averaged SST at $M_1$ (Figure 8a) and $M_2$ (Figure 8b) mooring locations from
observation and model forecasts. It should be noted that the observation at $M_2$ is available for a relatively shorter period from 21 October to 5 November 2020. Statistics of the SST bias, RMSD and correlation coefficient between the model forecast and the observations are listed in Table 5. The diurnal variability of SST at both locations is reasonably well reproduced by the model in all forecast lead times. However, the model forecasts have overestimated the SST diurnal variations from 23 October 2019. SST cooling during late October-early August 2020 at the location $M_1$ is underestimated in the model forecast. SST bias
and RMSD at $M_1$ are less than 0.07 ºC and 0.20 ºC, respectively, and remains fairly constant across all forecast lead times. Despite this, the correlation between model forecasts and observation depicts a considerable decrease at higher forecast lead





times. A cold SST bias of about -0.14 ºC to -0.17 ºC is noted at location $M_2$. The RMSD at $M_2$ is relatively less, with a maximum of 0.18 ºC across all forecast lead times while compared to $M_1$. The SST correlation at $M_2$ is above 95% confidence level ($r > 0.61$, $df \geq 9$) during all forecast lead times.

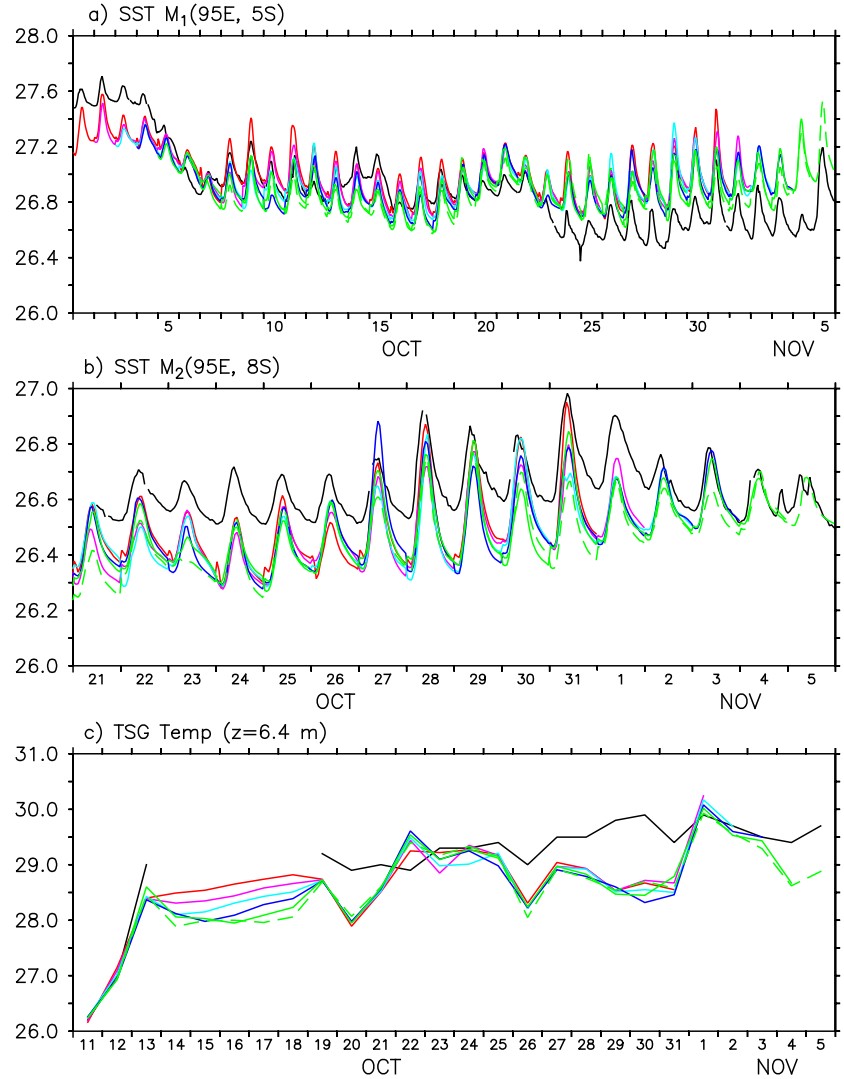

**Figure 8:** (a) and (b) Time series of hourly mean SST from model forecast and observations at the locations $M_1$ ($95^0$E, $5^0$S) and $M_2$ ($95^0$E, $8^0$S). Buoy locations are shown in Figure 3. (c) Subsurface temperature at 6.4 m depth for track shown in Figure 3 from TSG and model. Observation (black line), fcst_day1 (red line), fcst_day2 (purple line), fcst_day3 (light blue line), fcst_day4 (blue line), fcst_day5 (green line) and fcst_day6 (green dash).





(a)

| No | | Bias ($^0$C) | | | | | | RMSD ($^0$C) | | | | | |
|---|---|---|---|---|---|---|---|---|---|---|---|---|---|
| | | Forecast lead time (days) | | | | | | Forecast lead time (days) | | | | | |
| | | 1 | 2 | 3 | 4 | 5 | 6 | 1 | 2 | 3 | 4 | 5 | 6 |
| 1 | SST $M_1$ (95$^0$E, 5$^0$S) | 0.07 | 0.06 | 0.06 | 0.06 | 0.06 | 0.06 | 0.19 | 0.19 | 0.19 | 0.18 | 0.20 | 0.20 |
| 2 | SST $M_2$ (95$^0$E, 8$^0$S) | -0.14 | -0.17 | -0.17 | -0.15 | -0.15 | -0.16 | 0.15 | 0.18 | 0.18 | 0.17 | 0.17 | 0.18 |
| 3 | Temp (TSG, z = 6.4 m) | -0.47 | -0.43 | -0.43 | -0.43 | -0.41 | -0.46 | -0.65 | 0.65 | 0.66 | 0.68 | 0.64 | 0.64 |
| 4 | Temp $M_1$ (z = 0 - 600 m, 95$^0$E, 5$^0$S) | 1.87 | 1.87 | 1.88 | 1.91 | 1.91 | 1.95 | 2.83 | 2.82 | 2.83 | 2.88 | 2.89 | 2.96 |

(b)

| No | | Correlation Coefficient | | | | | |
|---|---|---|---|---|---|---|---|
| | | Forecast lead time (days) | | | | | |
| | | 1 | 2 | 3 | 4 | 5 | 6 |
| 1 | SST $M_1$ (95$^0$E, 5$^0$S) | 0.77 | 0.74 | 0.61 | 0.54 | 0.34 | 0.24 |
| 2 | SST $M_2$ (95$^0$E, 8$^0$S) | 0.94 | 0.91 | 0.83 | 0.79 | 0.78 | 0.68 |
| 3 | Temp (TSG, z = 6.4 m) | 0.87 | 0.86 | 0.85 | 0.83 | 0.84 | 0.86 |
| 4 | Temp $M_1$ (z = 0 - 600 m, 95$^0$E, 5$^0$S) | 0.60 | 0.61 | 0.58 | 0.55 | 0.52 | 0.56 |

**Table 5:** Items 1 and 2; Summary of SST bias and RMSD (a) and correlation coefficient (b) statistics between coupled ocean forecasts and observation at the mooring locations $M_1$ (95ºE, 5ºS) and $M_2$ (95ºE, 8ºS) during October 2019. Hourly averaged temperature from model and observation is used for the analysis. Item 3: Same as item 1 and 2, but for temperature at 6.4 m depth along the track shown in Figure 3. Daily averaged temperature from model and Instantaneous temperature at 1200 UTC from observation is used for the analysis. Item 4: Same as item 1 and 2, but for temperature within 0 to 600 m depth at the mooring location $M_1$. Daily averaged temperature model and observation s used in the analysis.


### 4.2.2 Temperature and salinity

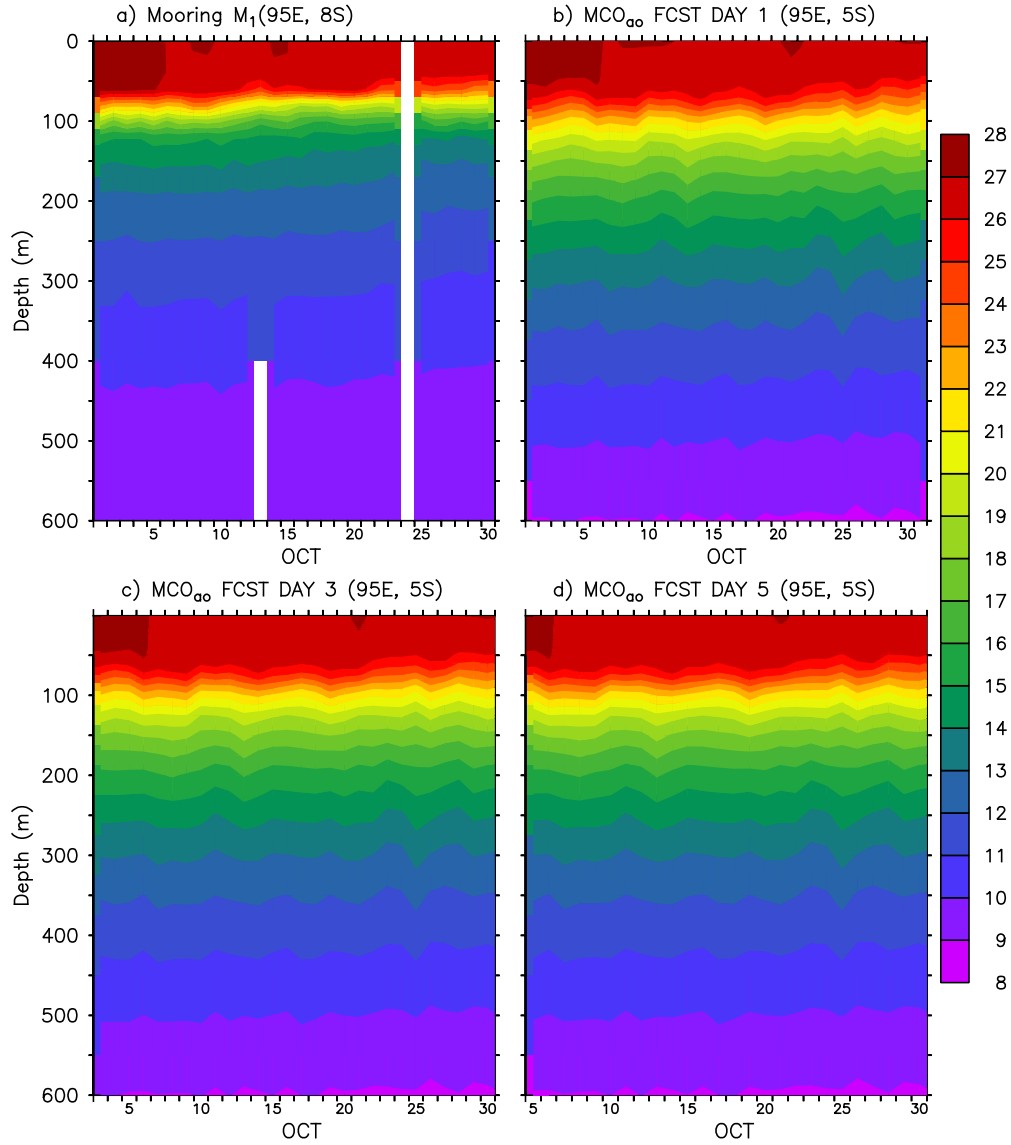

**Figure 9:** Depth-Time plots of temperature ($^0$C) at the $M_1$ mooring location from (a) buoy observation, and model forecast (b) fcst_day1, (c) fcst_day3 and (d) fcst_day5. x-axis starting date is different in (c) and (d).

Temperature and salinity data available from the CORIOLIS data portal is employed for the MCO$_{ao}$ forecast evaluation of subsurface fields. Argo and XBT profiles, moored buoys, and TSG observations are used for the analysis. We only consider mooring and TSG observations with a minimum of 10 days for the analysis. TSG observation selected for the analysis is

located in the Timor-Arafura Sea and the track has a direction of motion towards the west (Figure 3). Continuous observation





available at 6.4 m depth is compared with the model forecasts. Currently, the forecast system produces only daily averaged subsurface variables. Hence, the instantaneous TSG temperature observations at 1200 UTC are compared with the daily averaged model temperature. Time series of TSG temperature observation and model forecast for different forecast lead times is shown in Figure 8c. Though the model temperature shows a negative bias, daily temperature variation is reasonably well predicted by the model. Which is supported by high correlation, above 99.9% confidence level, between the model forecast and observation (Table 5, item no. 3). Largest temperature bias and RMSD are -0.47 ºC and 0.68 ºC, respectively, and significant variations in bias, RMSD and correlation statistics are not noticeable.

The moored buoy observations provide a unique opportunity to assess the model simulations both in the surface and subsurface of the ocean. Since the variables of interest in the present study are temperature and salinity, we have tried to compare the model forecast of temperature and salinity with buoy observation available at the locations $M_1$ and $M_2$. However, due to data gaps and shorter time series, salinity observation from both moorings and temperature from $M_2$ mooring are not included in the analysis. Depth-time plots of temperature from the model for forecast lead times 1 day (fcst_day1), 3 days (fcst_day3) and 5 days (fcst_day5), and moored observation, $M_1$, are shown in Figure 9. Statistics of temperature bias, RMSD and correlation coefficient between model forecast and observation are given in Table 5. The depth of the upper ocean isothermal/mixed layer and its shoaling in late October are well simulated by the model. A significant difference between the model forecast and observation is seen in the region below the mixed layer. The model simulation is unable to reproduce the sharp temperature stratification/cooling in the thermocline regions while this feature is well evident in the observation. This leads to relatively larger discrepancy between the model and observation in the subsurface region roughly between 100 m to 250 m depths. The usage of daily averaged temperature rather than instantaneous profile or higher vertical mixing in the model may be one of the possible reasons for this discrepancy. Larger temperature difference at the thermocline region has led to a warm temperature bias in the model forecast (Table 5). Maximum temperature bias and RMSD are 1.95 ºC and 2.96 ºC, respectively. Meanwhile, the correlation between the model forecast and observation is above 99% confidence level (r > 0.47) across all forecast lead times.

Argo and XBT profiles available for the period 1 October to 5 November 2019 are compared with model forecast to derive the RMSD statistics for temperature and salinity. Since no temperature or salinity profiles are available in the SCS (figure not shown, data distribution can be viewed from http://www.coriolis.eu.org/Data-Products/Data-Delivery/Data-selection), the analysis mainly demonstrates the model performance in the domain excluding the SCS region. As observed in the $M_1$ mooring location, warm biases with varying magnitude are seen, in the thermocline region, across the analysis-domain (figures not shown). Considering the depth range where this bias exists, the vertical mixing parameterization may have a strong influence in modifying the thermal stratification than the penetrative short wave forcing. Statistics of RMSD for ocean temperature and salinity relative to all profile observations is given in Table 6. RMSD of individual profiles are first computed and then rms value of the computed RMSD is derived. RMSD across all forecast lead times along with the number of profiles analyzed are listed. For both temperature and salinity, the RMSD remains fairly similar during the entire analysis period. Over the analysis-



domain and across all forecast lead times, the maximum RMSD for temperature and salinity are 1.41 ºC and 0.14 psu,
respectively. Overall, the model forecast deviation relative to observation is within acceptable error limits of operational
forecast models (e.g. Zhang et al., 2010; Yang et al., 2016).

| | RMSD | | | | | | |
|---|---|---|---|---|---|---|---|
| | Forecast lead time (days) | | | | | | All Forecasts |
| | 1 | 2 | 3 | 4 | 5 | 6 | |
| Temperature (ºC) 278 | 1.40 255 | 1.41 251 | 1.40 250 | 1.41 246 | 1.41 245 | 1.41 245 | 1.41 |
| Salinity (psu) 244 | 0.14 226 | 0.14 223 | 0.14 222 | 0.14 217 | 0.15 216 | 0.14 216 | 0.14 |

**Table 6:** Summary of temperature and salinity RMSD statistics between coupled ocean forecasts and in situ (Argo profile and
XBT) observations during October 2019. Daily averaged temperature from model and instantaneous temperature or salinity
from observation is used for the analysis. Red coloured number indicates the number of profiles analyzed for each variable
and lead forecast time.

### 4.2.3 Sea surface height

The same set of tide-gauge stations used for the MCO hindcast validation has been employed for MCO$_{ao}$ SSH forecast
evaluation. Due to irregularities in time series, the Currimao Ilocos Norte station is not included in the analysis. Hourly
instantaneous SSH data from the model forecast and tide-gauge observation is used for the statistical analysis. Summary of
SSH RMSD and bias statistics relative to the observations are listed in Table 7. Time series of hourly instantaneous SSH at
randomly selected tide-gauge stations (Sibolga, Prigi and Vung Tau) and MCO$_{ao}$ forecast (fcst_day1) are plotted in Figure 10.
Model SSH bias is within $\pm$0.10 m at all tide-gauge stations and forecast lead times. The SSH bias is within $\pm$ 0.05 m for 14
of total 19 tide-gauge stations across all forecast lead times. Since the tide-gauges are mostly located near to the coast, SSH
variability shorter than intra-seasonal timescale may be largely driven by the tidal forcing. Eventually, in the SSH bias, there
will be an offset between high and low tidal peaks. Hence, RMSD will give a better representation of model accuracy in tidal
dominant regions. RMSD above 0.15 m is observed at the tide-gauge stations in Hong Kong (0.18 m), Vung Tau (0.33 m), Ko
Lak (0.19 m), Ko Taphao Noi (0.17 m) and Pulau Langkawi (0.29 m). The SD of SSH observations at these stations during
October 2019 is 0.46 m, 0.82 m, 0.40 m, 0.71 m and 0.73 m, respectively, and the model forecast error is relatively lesser than
the observed SD. No significant variation in model forecast accuracy or RMSD is seen with the increase in forecast lead time.
The SSH RMSD is less than 0.10 m for 13 of the total 19 tide-gauge stations across all forecast lead times. Overall, the model
simulated SSH shows good agreement with the observations.



| No | Station name & Country | Latitude, Longitude | RMSD (m) | | | | | | Bias (m) | | | | | |
|---|---|---|---|---|---|---|---|---|---|---|---|---|---|---|
| | | | Forecast lead time (days) | | | | | | Forecast lead time (days) | | | | | |
| | | | 1 | 2 | 3 | 4 | 5 | 6 | 1 | 2 | 3 | 4 | 5 | 6 |
| 1 | Sabang, IDN | 5.888N, 95.317E | 0.06 | 0.06 | 0.06 | 0.06 | 0.06 | 0.07 | 0.01 | 0.01 | 0.01 | 0.01 | 0.02 | 0.02 |
| 2 | Sibolga, IDN | 1.75N, 98.767E | 0.07 | 0.06 | 0.06 | 0.07 | 0.07 | 0.08 | 0.01 | 0.02 | 0.03 | 0.04 | 0.04 | 0.05 |
| 3 | Padang, IDN | 1.0S, 100.367E | 0.05 | 0.05 | 0.05 | 0.05 | 0.05 | 0.05 | 0.00 | 0.00 | 0.00 | 0.00 | 0.00 | 0.00 |
| 4 | Cilicap, IDN | 7.752S, 109.017E | 0.09 | 0.08 | 0.08 | 0.08 | 0.08 | 0.08 | 0.04 | 0.04 | 0.04 | 0.04 | 0.04 | 0.04 |
| 5 | Prigi, IDN | 8.28S, 111.73E | 0.10 | 0.10 | 0.11 | 0.11 | 0.11 | 0.11 | 0.05 | 0.05 | 0.06 | 0.06 | 0.06 | 0.07 |
| 6 | Benoa, IDN | 8.745S, 115.21E | 0.11 | 0.10 | 0.10 | 0.10 | 0.10 | 0.10 | -0.03 | -0.03 | -0.02 | -0.02 | -0.02 | -0.02 |
| 7 | Saumlaki, IDN | 7.982S, 131.29E | 0.10 | 0.10 | 0.10 | 0.10 | 0.11 | 0.11 | 0.05 | 0.05 | 0.05 | 0.06 | 0.06 | 0.06 |
| 8 | Bitung, IDN | 1.44N, 125.193E | 0.10 | 0.10 | 0.10 | 0.10 | 0.10 | 0.10 | 0.08 | 0.08 | 0.08 | 0.08 | 0.08 | 0.08 |
| 9 | Malakal, PLW | 7.33N, 134.463E | 0.07 | 0.07 | 0.07 | 0.07 | 0.07 | 0.07 | 0.01 | 0.01 | 0.01 | 0.01 | 0.01 | 0.01 |
| 10 | Davao Gulf, PHL | 7.122N, 125.663E | 0.08 | 0.09 | 0.09 | 0.09 | 0.09 | 0.09 | -0.03 | -0.03 | -0.02 | -0.02 | -0.02 | -0.01 |
| 11 | Subic Bay, PHL | 14.765N, 120.252E | 0.04 | 0.04 | 0.04 | 0.03 | 0.03 | 0.03 | 0.00 | 0.00 | 0.00 | 0.00 | 0.00 | 0.00 |
| 12 | Manila, PHL | 14.585N, 120.968E | 0.08 | 0.08 | 0.07 | 0.07 | 0.07 | 0.07 | -0.04 | -0.04 | -0.04 | -0.04 | -0.04 | -0.04 |
| 13 | Legaspi, PHL | 13.15N, 123.75E | 0.06 | 0.06 | 0.06 | 0.06 | 0.06 | 0.06 | 0.01 | 0.01 | 0.01 | 0.01 | 0.01 | 0.01 |
| 14 | Hong Kong, HKG | 22.3N, 114.2E | 0.18 | 0.18 | 0.18 | 0.18 | 0.18 | 0.18 | -0.08 | -0.09 | -0.09 | -0.09 | -0.10 | -0.10 |
| 15 | Qui Nhon, VNM | 13.775N, 109.255E | 0.08 | 0.08 | 0.08 | 0.08 | 0.08 | 0.08 | -0.02 | -0.02 | -0.02 | -0.03 | -0.03 | -0.03 |
| 16 | Vung Tau, VNM | 10.34N, 107.072E | 0.33 | 0.33 | 0.32 | 0.33 | 0.32 | 0.32 | 0.00 | 0.00 | -0.01 | -0.01 | -0.02 | -0.02 |
| 17 | Ko Lak, THA | 11.795N, 99.817E | 0.17 | 0.16 | 0.16 | 0.19 | 0.18 | 0.19 | -0.03 | -0.02 | -0.04 | -0.07 | -0.07 | -0.08 |
| 18 | Ko Taphao Noi, THA | 7.832N, 98.425E | 0.17 | 0.17 | 0.17 | 0.17 | 0.17 | 0.17 | 0.00 | 0.00 | 0.00 | 0.00 | 0.00 | 0.01 |
| 19 | Pulau Langkawi, MYS | 6.432N, 99.765E | 0.29 | 0.27 | 0.27 | 0.26 | 0.26 | 0.25 | -0.05 | -0.04 | -0.03 | -0.03 | -0.03 | -0.02 |
| **Mean RMSD and Bias** | | | **0.14** | **0.14** | **0.14** | **0.14** | **0.14** | **0.14** | **0.00** | **0.00** | **0.00** | **0.00** | **0.00** | **0.00** |


**Table 7:** Summary of SSH RMSD and bias statistics between coupled ocean forecasts and tide-gauge observation during October 2019. Hourly instantaneous SSH from model and observation is used for the analysis.



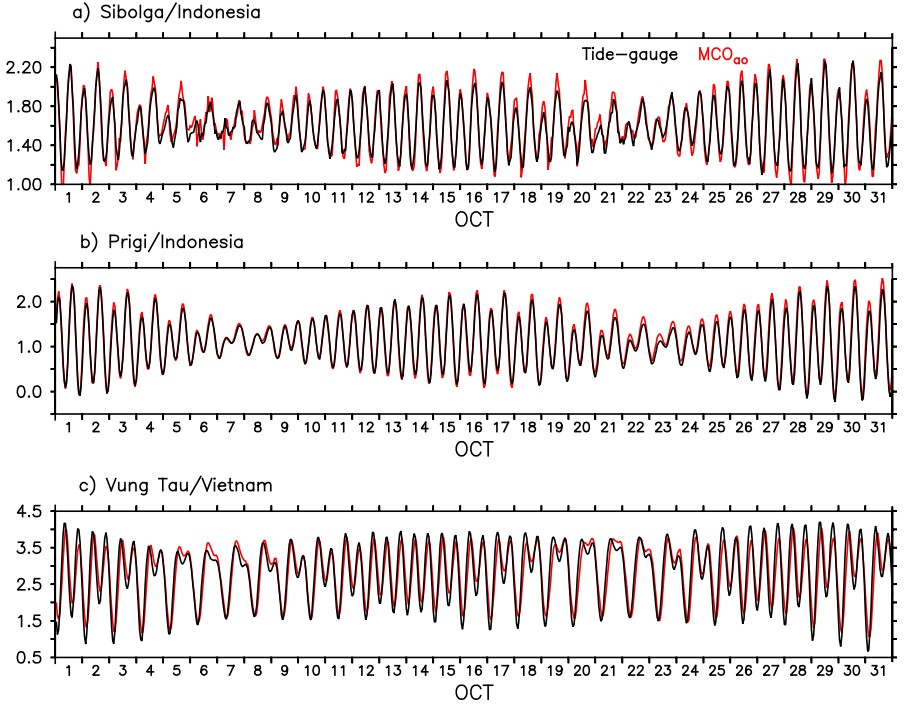

**Figure 10:** Time series of hourly instantaneous SSH (in m) from tide-gauge observation (black line) and $MCO_{ao}$ forecast lead time 1 day (fcst_day1, red line) at randomly selected stations (a) Sibolga ($1.75^0$N, $98.767^0$E), (b) Prigi ($8.28^0$S, $111.73^0$E)) and (c) Vung Tau ($10.34^0$N, $107.072^0$E) during October 2019.

## 5 Summary and future developments

The Maritime Continent has a profound influence on the global climate system because of its complex topography and unique geographic location within the tropical Indo-Pacific warm pool. The MC region is characterized by strong atmosphere-ocean coupled processes across multiple timescales. A convective-scale/eddy-resolving atmosphere coupled modelling system for the western MC, described in T18, was able to improve the simulation of a cold surge event, the intensity of Typhoon Sarika and its atmosphere-ocean interactions. Several upgrades has been added to the T18 model for a future operational

implementation, such as extending the eastern boundary of the model domain to the west Pacific Ocean, improved science configuration of the atmospheric model, MetUM, and incorporation of tidal boundary forcing to the ocean model, NEMO. Furthermore, the coupled model's feasibility to use as an operational forecast system is also being tested. Typical runtimes of the daily forecast simulations in our experiments are found to be suitable for the operational forecast applications.





The MC$_{ao}$ coupled prediction system has run as a pre-operational forecast system from 1 to 31 October 2019. Hindcast
simulations performed for the period 1 January 2014 to 30 September 2019, using the uncoupled ocean model MCO, has
provided the initial condition to the MCO$_{ao}$. The paper present details of atmosphere-ocean coupled prediction system
developed for the MC, and evaluations of ocean-only model hindcast and 6-day ocean forecast simulation performed using the
coupled system.

The evaluation of MCO hindcast is intended to understand the model's performance in reproducing the past ocean variability,
particularly at the ocean surface where the exchange of fluxes between the atmosphere and ocean takes place. The ERA5
driven simulations during the period from 1 January 2018 to 30 June 2019 is utilized for the evaluation of MCO. The SST
RMSD between model hindcast and OSTIA is less than 0.5 ºC for about 97% of the analysis-domain. Their correlation is
above 99.9% confidence level with about 88% of the analysis-domain displays correlation higher than 0.8. An SST cold bias
is seen in the Andaman Sea while most of the South China Sea, equatorial west Pacific Ocean and Australian coasts of the
Timor Sea show a warm bias. Overall, the mean SST bias, RMSD and mean correlation over the domain are 0.07 ºC, 0.34 ºC
and 0.90, respectively.

Comparison of model SST with the RAMA moored buoy observations located at the southeastern tropical Indian Ocean shows
a good agreement with each other. SST bias, RMSD and correlation coefficient between the model and observation are 0.17
ºC, 0.29 ºC and 0.94, respectively, for M$_1$ and 0.12 ºC, 0.41 ºC and 0.92, respectively, for M$_2$ locations. The SSH RMSD is
less than 0.10 m for 18 of total 20 tide-gauge stations analyzed and 14 stations show RMSD less than 0.05 m. Comparison of
model simulated SST and SSH fields show good agreement with observation and analysis data. Statistically significant
correlation with observation suggests that both the spatial and temporal patterns of variability are reasonably well reproduced
by the model.

For the evaluation of MCO$_{ao}$, comparisons of ocean forecast for different forecast lead times with OSTIA SST and in situ
observations have been performed. The forecasted SST over most of the sub-regions is within the error standard deviation of
the OSTIA. Though the model forecast exhibits a warm SST bias, the RMSD is less than 0.45 ºC over most of the sub-regions
during the analysis period. Generally, the forecasted SST tends to be cooler with an increase in forecast lead time, denoting a
lower warm bias and RMSD relative to fcst_day1. Overall, the SST correlation over the analysis-domain is above 99%
confidence level across all forecast lead times while the bias and RMSD are less than 0.19 ºC and 0.35 ºC, respectively.

The diurnal variability of SST at the RAMA moored buoy locations M$_1$ and M$_2$ are reasonably well reproduced by the model
across all forecast lead times. SST bias and RMSD at M$_1$ are less than 0.07 ºC and 0.20 ºC, respectively, and remains fairly
constant across all forecast lead times. Meanwhile, the RMSD at M$_2$ is relatively less, with a maximum of 0.18 ºC across all
forecast lead times. The depth of the upper ocean isothermal/mixed layer at the M$_1$ is well forecasted by the model. To
understand the model skill in predicting subsurface temperature and salinity, in situ profile observations are compared with





the model forecast for different forecast lead times. For both temperature and salinity, the RMSD remains fairly constant during the entire analysis period.

Comparison of model forecasted SSH shows good agreement with the tide-gauge observations. About 75% of the stations show bias within $\pm$ 0.05 m at all forecast lead times. No significant variation in model forecast accuracy or RMSD is seen with the increase in forecast lead time. About 73% of total stations show RMSD less than 0.10 m for all forecast lead times. Overall,

the model forecast deviation of SST, SSH, and subsurface temperature and salinity fields relative to observation is within acceptable error limits of operational forecast models (e.g. Zhang et al., 2010; Yang et al., 2016).

Analysis of subsurface fields revealed that significant model temperature biases exist in the region below the mixed layer. The representation of stratification in the thermocline region is relatively weak in the model forecast than the observations. This subsurface temperature bias remains across the model domain with varying magnitudes. Similar temperature biases are earlier

reported in simulations with identical model configurations (e.g. Graham et al., 2018). Further modelling works are needed to improve the thermal stratification through fine-tuning the mixing coefficients or modifying the vertical mixing parameterization.

Further analysis of model forecast fields using longer forecast simulations with increased observations will be undertaken to assess the model predictability across different seasons and during typical weather events such as cold surge, typhoon, MJO.

In addition, the impact of coupling on the forecast will be investigated by performing simultaneous stand-alone ocean model forecast simulations. Further, works are ongoing to understand the forecast skill of $MCA_{ao}$ and results from the analysis will be presented as research publications.

The evaluation of sea surface salinity and ocean current fields are not included in the present study mainly due to the lack of in situ and satellite observations. Especially, the South China Sea remains as a data-sparse region in terms of the subsurface

observations that highlights the necessity of coordinated efforts from the scientific community to fill these spatial data gaps.

In our analyses, the RMSD and positive or negative biases of the ocean forecasts are generally comparable to those observed from the hindcast statistics. This suggests that up to a certain extent, the model forecast deviation is inherited from the MCA hindcast or the $MCA_{ao}$ initial condition. The dependency of model forecast quality on the initial state is well established in the numerical weather prediction studies. In particular, over the tropical oceans, the initialization of the ocean state is an important

element of the forecast systems. The data assimilation techniques help to acquire an improved estimate of the ocean state by combining the model simulated fields and observations (King et al., 2018). Both the uncoupled and coupled ocean configurations used in our study are free-running models with no restoring or relaxation to the real world. Hence, to provide a better initial condition to $MCO_{ao}$, implementation of data assimilation capability to MCO will have a key priority in our future developments. Also, earlier studies have shown that including wave-induced mixing in ocean circulation model yields a better

representation of the upper ocean temperature (Lewis et al., 2019b). Thus, work towards the development of a two-way atmosphere-ocean-wave coupled system will be undertaken in the future.

**Model code and data availability:**

Due to intellectual property right restrictions, the coupled model system code cannot be provided directly. However, full source codes, scripts and related forcing datasets used in the study can be made available to the Editor for review. The Met Office Unified Model (MetUM) is available for use under licence from UK Met Office via a shared MetUM code repository, which can be accessed via https://code.metoffice.gov.uk/trac/um/wiki. The NEMO vn3.6 model code is freely available from the NEMO website (https://www.nemo-ocean.eu). The OASIS3-MCT coupler is disseminated to registered users as free software from https://verc.enes.org/oasis. All model outputs analyzed in the manuscript can be made available upon contacting the authors.

**Competing interests:**

The authors declare that they have no conflict of interest.

**Author contributions:**

BT and CS developed the coupled modelling system. BT performed model simulations, analysed data and wrote the paper with inputs from CS. BCPH and RK contributed to the system development. JL provided system and software support to the study. XYH and PT performed Funding acquisition and provided supervision. BCPH, RK, JL, XYH, PT contributed to the discussion and improvement of the paper.

**Acknowledgements:**

The model simulations are performed in the Cray XC-30 HPC system housed at CCRS. We acknowledge Dr. Christopher Gordon, Dr. Huw Lewis, Dr. Enda O'Dea, Dr. Juan Manuel Castillo and Dr. Jeniffer Graham for their scientific and technical support. We acknowledge ECMWF, Copernicus Marine Environment Monitoring Service, Coriolis, UHSLC and aviso for various datasets used in the study. Figures are drawn using ferret and python softwares.

**Financial Support:**

The project is funded by the National Environmental Agency, Singapore through its Meteorological Service Singapore as a collaborative research between Centre for Climate Research Singapore (CCRS) and National University of Singapore (NUS).





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
