# Peer review of "Development of a MetUM (v 11.1) – NEMO (v 3.6) coupled operational forecast model for the Maritime Continent: Part 1 - Evaluation of ocean forecasts"

_Geoscientific Model Development, 2020_

## Short Comment (SC1) · 22 Oct 2020

Thank you for the discussion about coupling in operational forecasts.

However, it was not clear to me what the improvements are in having the coupled system and at what cost (computationally). I believe it would be great to have this information highlighted in the summary.

For the Argo profiles, it would be nice to have the statistics by depth range (eg. mixed layer, thermocline, and ocean interior), and displayed in a map with MCO and MCOao

side by side, to identify regions with larger improvements.

Cheers

---

## Referee Comment (RC1) · Anonymous Referee #1 · 4 Nov 2020

**General comments**

The authors start by presenting the development of an atmosphere-ocean coupled operational forecast model for the Maritime Continent. The authors then evaluate the model's performance to forecast SST, temperature and salinity in the water column, and SSH. The coupled model uses OASIS3-MCT libraries to couple a state-of-the-art atmospheric (MetUM) and ocean (NEMO) model with a 4.5x4.5 km horizontal resolution. Besides, the authors evaluate the ocean-only model (NEMO) in hindcast mode.

[Figure]

Overall, the paper addresses relevant scientific modeling questions within the scope of the Geoscientific Model Development (GMD). The development of a 4.5x4.5 km horizontal resolution ocean model coupled with an atmospheric model for the complex environment of the Maritime Continent can, in the future, be used to address relevant scientific questions within the scope of the European Geosciences Union (EGU).

The paper represents advances in ocean modeling for the Maritime Continent, and its methodology can be adapted to other oceans. This is because the methods and assumptions are valid and clearly outlined by the authors. Overall, the results presented for hindcast and forecast simulations are sufficient to support the authors' interpretations and conclusions. The paper is well structured; the abstract provides a concise and complete summary of the work presented. The language is fluent and precise. Overall, the number and quality of references are appropriate; however, some references are missing in the references list.

Based on this, my opinion is that GMS should publish this paper. However, some contents deserve to be better presented and discussed. The following sections present all the comments and recommendations following my review.

Specific comments

Comment #1 – (Figure 1 and line 149) A reference for GEBCO 2014 should be provided and added to the references list. Are the authors considering updating MCao bathymetry based on the GEBCO 2020 dataset? Although GEBCO 2019 and GEBCO 2020 were not available when the simulations were performed, these datasets were available when the paper was submitted. The authors should then discuss how using a 15 arc-second resolution bathymetry dataset could influence model results.

Comment #2 – (line 80 and 81) Please provide references to support this statement ("For instance,... Pacific Ocean").

Comment #3 - (line 81 and 82) Please provide references to support this statement

("The north Pacific Ocean... tropical cyclone annually").

Comment #4 – (line 116 and 117) Why did the authors decide to change the values of background vertical eddy viscosity and eddy diffusivity coefficients?

Comment #5 – (line 167) Could the authors briefly explain how they managed not to have these numerical issues in MCao? This information can be handy for other authors that want to implement similar model systems based on NEMO.

Comment #6 - (line 168) FES2014b is not in the list of references.

Comment #7 - (line 190) What is the external source for MSLP?

Comment #8 – (line 194) Please provide a reference for Mercator global ocean reanalysis.

Comment #9 – (line 254) Please provide a reference for the Operational Sea Surface Temperature and Sea Ice Analysis.

Comment #10 – (Figure 4): Although Figure 3 mentioned that the Bay of Bengal region is excluded from the analysis-domain, results are presented for this region in Figure 4. Could the authors clarify this better in the text? Moreover, I suggest writing the abbreviations of each sub-region in the map of figure 3. This will help readers not familiarized themselves with the Maritime Continent.

Comment #11 (line 295) – What could be the reason for the observed SST bias in the Andaman Sea region?

Comment #12 (line 366 to line 368) – What could explain this?

Comment #13 (Table 4) – Can the author elaborate on the reasons for the decrease of SST Bias overtime (Bias decreases for higher forecast lead times). This is a general trend for all the sub-regions (exception for ASMS). In general, it is expected that the accuracy of SST decreases with increasing lead times. However, it seems that Bias is not showing that.

[Figure]

Comment #14 - In Figure 9a, results are presented for Mooring M1(95E, 8S). However, in Figure 9b, 9c and 9d results are presented by MCO_ao at 95E, 5S. Is this a typo? Based on the text in the manuscript, I believe this is a typo. However, the authors must double-check if they compared observations and model results at the same lat/lon.

Comment #15 (line 424 to line 435, and Table 6) – The authours say "the analysis mainly demonstrates the model performance in the domain excluding the SCS region". This is understandable. However, it is impossible to understand if Argo and XBT profiles available from 1 October to 5 November 2019 represent the other sub-regions. Did the author calculate RMSD for all the sub-regions (excluding SCS)? Does RMSD change from sub-region to sub-region?

Comment #16 (Table 6) The legend of Table 6 indicates "observation during October 2019". However, in line 424 the authors mentioned: "for the period 1 October to 5 November". Please clarify this point.

Comment #17 (table 7) - The 19 tide-gauge location used to evaluate the model perfor-mance to simulate SSH should be presented in Figure 3. Although the lat/lon of each tide-gauge is provided in Table 7, its location in a map will make it easier for readers to understand better where the authors evaluated SSH performance.

Technical corrections

Line 117 – In line 111 MC_ao is defined as MCao. Please use only one nomenclature for the MC atmospheric-ocean coupled model.

Line 143 – Do you mean Parallelise instead of PArallelise?

Line 269 – Replace "set of variables an d" by "set of variables and"

Table 5 – Please delete the space in "observation s"

---

## Short Comment (SC2) · 14 Nov 2020

Dear authors,

in my role as Executive editor of GMD, I would like to bring to your attention our Editorial version 1.2:

https://www.geosci-model-dev.net/12/2215/2019/

This highlights some requirements of papers published in GMD, which is also available

on the GMD website in the 'Manuscript Types' section:

http://www.geoscientific-model-development.net/submission/manuscript_types.html

In particular, please note that for your paper, the following requirement has not been met in the Discussions paper:

- "The main paper must give the model name and version number (or other unique identifier) in the title."

Please add the names of the coupled models and their version numbers or name and version number of the coupled system to the title of your article upon paper revision.

Yours,

Astrid Kerkweg

---

## Referee Comment (RC2) · Anonymous Referee #2 · 23 Nov 2020

This paper is an important contribution in the area of regional coupled model development. Authors have analyzed the ocean forecasts from a regional coupled suite which is under development. Team of Scientists/researches from different part of the world are working to develop a suitable regional coupled model which can be used to zoom-in (run at a very high resolution) for any extreme event or case such as depression/front or tropical cyclone forecasting. SST Bias and RMSD and correlations are very promising from this model over the oceanic part of the domain of interest. Times series of the domain averaged SST drifts shown by authors over the period of a year is

an interesting result as well, however, this needs further work and probably beyond the scope of this paper. I suggest authors to also look in to the atmospheric part (which I am sure they must be planning) and try to find answer for this SST drift. Hindcast SST's are in agreement with those of RAMA moorings, tide-gauge also is very close to the observation. SST bias of less than 0.2 in 6 day lead forecast is a significant result for such a high resolution forecast. Mean correlation of 0.5 for 6 day lead forecast is also a very good number which is 0.85 for Java Sea. Vertical thermal structure of the forecasted ocean looks very promising up to day 3. SSH results are also in well agreement with observation. Overall this is a great work and should get publish. I have been reading some suggested typos correction by other reviewer which I don't want to repeat and believe that if those typing mistakes are corrected this manuscript is worthy for publication.

―――――――――――――――――――――

---

## Referee Comment (RC3) · Anonymous Referee #3 · 24 Nov 2020

The manuscript presents the development of an atmosphere-ocean coupled operational forecast model for the maritime continent at 4.5 x 4.5 km horizontal resolution and the evaluation of ocean-only model hindcast and operational forecast simulations. Regional settings of MetUM and NEMO models are coupled using the OASIS3-MCT software. The manuscript first presents the details of the atmospheric and ocean models, and further discuss the coupling methodology as well as the operational forecast system settings. To assess the NEMO performance, the model hindcast and forecast simulations are compared with the observations of SST, water column temperature

and salinity, and the SSH. The manuscript is well written and the results are presented clearly. Development of a regional coupled ocean-atmosphere model is a tedious task and it requires all kind of expertise manpower and support from funding agencies. This manuscript represents a substantial contribution to the modelling studies focusing on the Maritime Continent regions and it is within the scope of the journal GMD. On the scientific merit, this work is an excellent peace of quality research and I recommend to publish it in GMD with a few minor corrections. My specific comments are given below.

1. Figure 1: Improve readability/increase font size of figure 1 labels. 2. L 190: mention the mslp data source 3. L192: Any reason for "69-month" ocean hindcast run? 4. Fig 3: Provide sub-region abbreviations inside the figure hence the readability can be improved. 5. Table 3: It will be appropriate to provide the tide-gauge locations in the figure 3. 6. Figure 9a. Looks a typo in figure 9a M1 location. 7. L521: please provide a brief outlook on the analysis which will be included in the MCAao . 8. L587: Update the reference, if available.

---

## Author Comment (AC1) · 9 Dec 2020

Thanks for your kind interest in our discussion paper and for the constructive comments. Our response to the short comments (SC) are given in bold font.

SC: However, it was not clear to me what the improvements are in having the coupled system and at what cost (computationally). I believe it would be great to have this information highlighted in the summary.

**Preliminary aim of the present study was to develop the coupled modeling system for a larger domain of the Maritime Continent, than Thompson et al (2018, climate dynamics), with upgraded model settings and to make system ready for operational forecast application. Hence, at present no stand-alone NEMO ocean forecast system for the domain has been configured. The MCO simulations in our study represents only the stand-alone NEMO hindcast (or T0 minus 1 day) runs. Thus, we have presented the comparison of coupled ocean forecast simulations with observational data sets only in the discussion. However, as part of our future studies we have plan to develop stand-alone ocean forecast configuration and to compare the coupled and uncoupled model simulations to understand the impact of coupling. We have provided this info in the summary section of the discussion.**

**Meanwhile, our ongoing work with the atmospheric forecast evaluation will mainly include the comparison of the coupled against atmosphere-only forecast simulations over the same region. Regarding the computational cost, the coupled model (np # 1008) need only about 16% more processors than the uncoupled atmosphere model (np # 864) (Table 1).**

SC: For the Argo profiles, it would be nice to have the statistics by depth range (eg. mixed layer, thermocline, and ocean interior), and displayed in a map with MCO and MCOao C1 side by side, to identify regions with larger improvements.

**In the analysis, we have identified that the region below mixed layer or the thermocline region is noted by significant temperature biases. Our preliminary analysis showed that the RMSD in the mixed layer, thermocline and ocean interior region are within the range 0 - .5 $^o$C, 0.5 - 4.5 $^o$C and 0 - .2 $^o$C, respectively for the coupled forecast simulations during the study period. This remains almost uniform over the model domain. But this result is based on the analysis of randomly selected profiles from different sub regions. As mentioned above, since our study doesn't involve MCO forecast simulations, the comparisons of MCO and MCO$_{ao}$ are not included in the present study. However, as part of our future studies more detailed analysis of model deviation from observations at the subsurface region at different sub-regions and the impact of coupling will be preformed.**

---

## Author Comment (AC2) · 9 Dec 2020

As suggested by the executive editor we have modified the discussion manuscript title. The revised manuscript title will be "Development of a MetUM (v 11.1) – NEMO (v 3.6) coupled operational forecast model for the Maritime Continent: Part 1 - Evaluation of ocean forecasts"

---

## Author Comment (AC3) · 15 Dec 2020

**We thank the reviewer for his/her careful reading and constructive comments which helped to improve the manuscript (ms). The point-by-point replies to the reviewer's comments are provided below in bold font:**

Specific comments

Comment #1 – (Figure 1 and line 149) A reference for GEBCO 2014 should be provided and added to the references list. Are the authors considering updating MCao bathymetry based on the GEBCO 2020 dataset? Although GEBCO 2019 and GEBCO 2020 were not available when the simulations were performed, these datasets were available when the paper was submitted. The authors should then discuss how using a 15 arc-second resolution bathymetry dataset could influence model results.

**Web link to the GEBCO 2014 data is included in the revised manuscript. In our future developments of the coupled system, the latest available bathymetric data sets will be used. More accurate representation of the model bathymetry is expected to improve the simulations of tide sea level and also the ocean circulation patterns**.

Comment #2 – (line 80 and 81) Please provide references to support this statement ("For instance,. . . Pacific Ocean").

**Reference included in the revised ms (L80).**

Comment #3 - (line 81 and 82) Please provide references to support this statement ("The north Pacific Ocean. . . tropical cyclone annually").

**Reference included in the revised ms (L82).**

 Comment #4 – (line 156 and 157) Why did the authors decide to change the values of background vertical eddy viscosity and eddy diffusivity coefficients?

**We have conducted a few sensitivity experiments and found that reduced background viscosity and diffusivity coefficients has led to an improvement in the model simulations at the model resolution we used. Hence, those reduced values are used in our model simulations.**

Comment #5 – (line 167) Could the authors briefly explain how they managed not to have these numerical issues in MCao? This information can be handy for other authors that want to implement similar model systems based on NEMO.

**A few steps were taken to overcome this issue from T18 to MCAao. Mainly we have adjusted the bathymetry manually by comparing gebco with the navigation charts.**

Comment #6 - (line 168) FES2014b is not in the list of references.

**Reference added (L168).**

Comment #7 - (line 190) What is the external source for MSLP?

**ECMWF IFS. A sentence added in the revised ms (L189-190).**

Comment #8 – (line 194) Please provide a reference for Mercator global ocean reanalysis.

**Reference added to the data discussion in section 3.1 (L239).**

Comment #9 – (line 254) Please provide a reference for the Operational Sea Surface Temperature and Sea Ice Analysis.

**Reference included in the data discussion in section 3.1 (L258).**

Comment #10 – (Figure 4): Although Figure 3 mentioned that the Bay of Bengal region is excluded from the analysis-domain, results are presented for this region in Figure 4. Could the authors clarify this better in the text? Moreover, I suggest writing the abbreviations of each sub-region in the map of figure 3. This will help readers not familiarized themselves with the Maritime Continent.

**In the figure caption, the sentence modified as "The Bay of Bengal region (north of 5ºN, west of 92ºE) is excluded when analysis is performed for different sub-regions."**

**The abbreviations are included in figure 3.**

Comment #11 (line 295) – What could be the reason for the observed SST bias in the Andaman Sea region?

**From the present study, we are not able to completely understand the reasons behind this SST drift. The possible reasons for this SST drift may include the effects of wave-induced mixing, the air-sea heat fluxes and surface wind patterns, etc. More detailed analyses are required to identify the factors responsible for this discrepancy.**

Comment #12 (line 366 to line 368) – What could explain this?

Comment #13 (Table 4) – Can the author elaborate on the reasons for the decrease of SST Bias overtime (Bias decreases for higher forecast lead times). This is a general trend for all the sub-regions (exception for ASMS). In general, it is expected that the accuracy of SST decreases with increasing lead times. However, it seems that Bias is not showing that.

**Comment 12 -13: As pointed out the reviewer, it is expected that the quality of model simulations decreases with higher forecast lead times. The decrease in SST correlation with increase in lead times is in accordance with the above notion. Meanwhile, decrease in the SST bias or RMSD as seen in our study is not consistent with the above discussion. Since the present study focuses mainly in the development of an operational coupled modeling system and evaluation of ocean forecast fields, detailed analysis to isolate the mechanisms behind the drifts in forecast fields are not undertaken. Our ongoing atmospheric forecast analysis may provide useful insights in identifying the factors contribute to such variations/discrepancies.**

Comment #14 - In Figure 9a, results are presented for Mooring M1(95E, 8S). However, in Figure 9b, 9c and 9d results are presented by MCO_ao at 95E, 5S. Is this a typo? Based on the text in the manuscript, I believe this is a typo. However, the authors must double-check if they compared observations and model results at the same lat/lon.

**Corrected the M₁ location as 95E, 5S.**

Comment #15 (line 424 to line 435, and Table 6) – The authours say "the analysis mainly demonstrates the model performance in the domain excluding the SCS region". This is understandable. However, it is impossible to understand if Argo and XBT profiles available from 1 October to 5 November 2019 represent the other sub-regions. Did the author calculate RMSD for all the sub-regions (excluding SCS)? Does RMSD change from sub-region to sub-region?

[Figure]

**Number of observations available during 01/10/2019 to 05/112019/. Argo, XBT and CTD profiles are used for the subsurface temperature and salinity analysis.**

**The Argo/XBT/CTD profiles used for the subsurface temperature/salinity forecast evaluation are shown in the figure. As seen, the profiles are mostly confined to the Tropical eastern Indian Ocean and tropical western Pacific Ocean. A few profiles are present in the Timor-Arafura and Banda Sea regions. The temperature and salinity RMSD variation between these sub-regions are less than 0.3 deg C and 0.025 psu, respectively.**

Comment #16 (Table 6) The legend of Table 6 indicates "observation during October 2019". However, in line 424 the authors mentioned: "for the period 1 October to 5 November". Please clarify this point.

**Corrected as "Summary of temperature and salinity RMSD statistics between coupled ocean forecasts and in situ (Argo profile and XBT) observations during 1 October to 5 November 2019."**

Comment #17 (table 7) - The 19 tide-gauge location used to evaluate the model performance to simulate SSH should be presented in Figure 3. Although the lat/lon of each tide-gauge is provided in Table 7, its location in a map will make it easier for readers to understand better where the authors evaluated SSH performance.

Technical corrections

Line 117 – In line 111 MC_ao is defined as MCao. Please use only one nomenclature for the MC atmospheric-ocean coupled model.

**Corrected**

Line 143 – Do you mean Parallelise instead of PArallelise?

**We follow the naming convention used for the OPA model, hence it remains as PArallelise**

Line 269 – Replace "set of variables an d" by "set of variables and"

**Corrected**

Table 5 – Please delete the space in "observation s"

**Corrected**

---

## Author Comment (AC4) · 15 Dec 2020

**We thank the reviewer for his/her careful reading and constructive comments which helped to improve the manuscript (ms). The point-by-point replies to the reviewer's comments are provided below in bold font:**

Specific comments

1. Figure 1: Improve readability/increase font size of figure 1 labels.

**Improved figure presented in the revised version**

2. L 190: mention the mslp data source

**ECMWF IFS. A sentence added in the revised ms (L189-190).**

3. L192: Any reason for "69-month" ocean hindcast run?

**We have started the model from Jan 2014 to have spin-up period of 4 years (2014-2017) and atleast 1 year (2018) of simulation for the hindcast validation. Due to the delay in bringing the coupled system to a pre-operational forecast stage we have extended the hindcast simulations to Sep 2019. This results in total 69 months of hindcast run.**

4.Fig 3: Provide sub-region abbreviations inside the figure hence the readability can be improved.

**The abbreviations are included in the revised figure.**

5. Table 3: It will be appropriate to provide the tide-gauge locations in the figure 3.

**The tide gauge locations are included in the revised figure**

6. Figure 9a. Looks a typo in figure 9a M1 location.

**Corrected in the revised manuscript.**

7. L521: please provide a brief outlook on the analysis which will be included in the MCAao.

**Included in the revised manuscript (L520-522).**

8. L587: Update the reference, if available.
**Reference is updated.**

---

## Author Comment (AC6) · 15 Dec 2020

We thank the reviewer for his/her careful reading and encouraging comments on our manuscript (ms). We are currently working on the atmospheric forecast evaluation and to identify the factors responsible for the SST forecast drift. We tried our best to address the comments by the other two reviewers.

---

## Author Response (AR2)

**Author Response to the comments by Topical Editor**

**We thank the Editor for considering our manuscript (ms) for possible publication in GMD. Our reply to the Editor's comments is provided below in bold font:**

The authors have sufficiently addressed reviewer comments. The authors need to ensure that further checks are done when uploading a final copy. This includes proper documentation of open source code and data.

**We have carefully read the ms. and corrected typos and formatted the ms. (figures, references, etc.) according to GMD requirements. We have modified the 'model code and data availability' section to address the Editor's comments. The revised section includes the discussion on how to obtain the model source codes and data, and the technical details of the open source NEMO model. The revised section from the ms. is pasted below.**

*"*

**Due to intellectual property right restrictions, the coupled model system, MetUM or JULES source codes and documentations cannot be provided directly. However, full source codes and scripts used in the study can be made available to the Editor for review.**

*Obtaining the MetUM.* **The Met Office Unified Model is available for use under license from UK Met Office via a shared MetUM code repository, which can be accessed via https://code.metoffice.gov.uk/trac/um/wiki. A number of research organizations and national meteorological services use the UM in collaboration with the Met Office to undertake basic atmospheric process research, produce forecasts, develop the UM code, and build and evaluate Earth system models. For further information on how to apply for a license, see http://www.metoffice.gov.uk/research/modelling-systems/unified-model.**

*Obtaining JULES.* **JULES is available under license free of charge. For further information on how to gain permission to use JULES for research purposes, see http://jules-lsm.github.io/access_req/JULES_access.html.**

*Obtaining NEMO.* **The NEMO vn3.6 model code and documentation is freely available from the NEMO website (https://www.nemo-ocean.eu). The details of NEMO branches, compilation keys and namelist parameters used in our modeling systems are described in the supplement.**

*Obtaining OASIS-MCT.* **The OASIS3-MCT coupler is disseminated to registered users as free software from https://verc.enes.org/oasis.**

*Data.* **The data size of coupled forecast model output in our simulations are of several terabytes and requires large storage facility. However, all model outputs analyzed in the manuscript can be made available upon contacting the authors. The observational datasets used for the model evaluation are freely available and data sources are described in the data discussion section of the manuscript.**

*"*